

# Mid/Late Devonian-Carboniferous collapse basins on the Finnmark Platform and in the southwesternmost Nordkapp basin, SW Barents Sea

5  Jean-Baptiste Koehl[1,2], Steffen G. Bergh[1,2], Tormod Henningsen[1], Jan-Inge Faleide[2,3]

[1]Department of Geosciences, University of Tromsø, N-9037 Tromsø, Norway.

[2]Research Centre for Arctic Petroleum Exploration (ARCEx), University of Tromsø, N-9037 Tromsø, Norway.

[3]Department of Geosciences, University of Oslo, P.O. Box 1047 Blindern, NO-0316 Oslo, Norway.

*Correspondence to*: Jean-Baptiste Koehl (jean-baptiste.koehl@uit.no)

**Abstract**. The SW Barents Sea margin experienced a pulse of extensional deformation in the Middle-Late Devonian through the Carboniferous, after the Caledonian Orogeny terminated. These events marked the initial stages of formation of major offshore basins such as the Hammerfest and Nordkapp basins. We mapped and analyzed three major fault complexes, i) the Måsøy Fault Complex, ii) the Rolvsøya fault, iii) the Troms-Finnmark Fault Complex. We discuss the formation of the Måsøy Fault Complex as a possible extensional splay of an overall NE-SW trending, NW-dipping, basement-seated Caledonian shear zone, the Sørøya-Ingøya shear zone, which was partly inverted during the collapse of the Caledonides and accommodated top-to-the-NW normal displacement in Mid/Late Devonian-Carboniferous times. The Troms-Finnmark Fault Complex displays a zigzag-shaped pattern of NNE-SSW and ENE-WSW trending extensional faults before it terminates to the north as a WNW-ESE trending, NE-dipping normal fault that separates the southwesternmost Nordkapp basin in the northeast from the Finnmark Platform west and the Gjesvær Low in the southwest. The WNW-ESE trending, margin-oblique segment of the Troms-Finnmark Fault Complex is considered to represent the offshore prolongation of a major Neoproterozoic fault complex, the Trollfjord-Komagelv Fault Zone, which is made of WNW-ESE trending, subvertical faults that crop out on the island of Magerøya in NW Finnmark. Our results suggest that the Trollfjord-Komagelv Fault Zone dies out to the northwest before reaching the Finnmark Platform west. We propose an alternative model for the origin of the WNW-ESE trending fault segment of the Troms-Finnmark Fault Complex as a possible hard-linked, accommodation cross-fault that developed along the Sørøy-Ingøya shear zone. This brittle fault



decoupled the Finnmark Platform west from the southwesternmost Nordkapp basin and merged with the Måsøy Fault Complex in Carboniferous times. Seismic data over the Gjesvær Low and southwesternmost Nordkapp basin show that the low-gravity anomaly observed in these areas may result from the presence of Mid/Late Devonian sedimentary units resembling Middle Devonian, spoon-shaped, late/post-orogenic collapse basins in western and mid Norway. We propose a model for the formation of the southwesternmost Nordkapp basin and its counterpart Devonian basin in the Gjesvær Low by exhumation of narrow, ENE-WSW to NE-SW trending basement ridges along a bowed portion of the Sørøya-Ingøya shear zone in the Mid/Late Devonian-early Carboniferous. Exhumation may have involved part of a large-scale metamorphic core complex that potentially included the Lofoten Ridge, the West Troms Basement Complex and the Norsel High. Finally, we argue that the Sørøya-Ingøya shear zone truncated and decapitated the Trollfjord-Komagelv Fault Zone during the Caledonian Orogeny and that the western continuation of the Trollfjord-Komagelv Fault Zone was mostly eroded and potentially partly preserved in basement highs in the SW Barents Sea.

## 1. Introduction

The SW Barents Sea margin is located near the Iapetus suture zone that formed when Laurentia collided with Fennoscandia to produce the Caledonian Orogeny (Ramberg et al., 2008; Gernigon et al., 2014). This suture and possible, related deep-seated shear zones accommodating e.g. thrust nappe emplacement during the Caledonian Orogeny are now covered by late Paleozoic to Cenozoic sedimentary basins that formed during multiple episodes of extension. These repeated extension events led to the breakup of the North Atlantic Ocean and formation of a transform plate margin at the boundary between the Mid-Norwegian and SW Barents Sea margins (Faleide et al., 1993, 2008; Blystad et al., 1995; Doré et al., 1997; Bergh et al., 2007; Hansen et al., 2012; Gernigon et al., 2014). The rift-margin along the SW Barents Sea, offshore Western Troms and NW Finnmark (Figure 1), consists of the Finnmark Platform and an adjacent, glacial sediment-free strandflat, and of deep offshore basins such as the Hammerfest and Nordkapp basins (Gabrielsen et al., 1990). These basins are bounded by major NE-SW trending extensional faults such as the Troms-Finnmark Fault Complex (TFFC; Gabrielsen et al., 1990; Smelror et al., 2009; Indrevær et al., 2013), the Måsøy Fault Complex (MFC; Gabrielsen et al., 1990; Gudlaugsson et al., 1998), and potential basement-seated ductile detachments. The study area also includes a deep Paleozoic basin



that is located southwest of the Nordkapp Basin and east of the Hammerfest Basin, and which is bounded to the southwest by the WNW-ESE trending segment of the TFFC and to the southeast by the MFC (Figure 1). This basin was named the "easternmost Hammerfest basin" by Omosanya

et al. (2015). We find this name inappropriate and tentatively rename this basin the "southwesternmost Nordkapp basin", as argued for later in the text.

In addition, the SW Barents Sea margin off Western Troms and NW Finnmark is segmented by NW-SE trending transfer fault zones, e.g. Senja Shear Zone and Fugløya transfer zone (Indrevær et al., 2013), which are both sub-parallel to the onshore, Neoproterozoic, WNW-ESE trending

Trollfjord-Komagelv Fault Zone (TKFZ) in eastern Finnmark (Siedlecki, 1980; Herrevold et al., 2009) and to the Kokelv Fault on the Porsanger Peninsula (Figure 1; Gayer et al., 1985; Lippard & Roberts, 1987; Rice, 2013) while the coastal Langfjord-Vargsund fault (LVF) trends NE-SW, parallel to the TFFC (Figure 1). The TKFZ is believed to continue farther west, off the coast, where it is thought to interact with and merge into the WNW-ESE trending fault segment of the TFFC

(Gabrielsen, 1984; Vorren et al., 1986; Towsend, 1987; Gabrielsen & Færseth, 1989; Gabrielsen et al., 1990; Roberts et al., 2011; Bergø, 2015; Lea, 2015). The geometric interaction, timing and controlling effects of the TFFC, MFC, TKFZ and LVF, and adjacent offshore basins and ridges are not yet resolved. In particular, the presence of potential Caledonian structures in the deeper portion of the Finnmark Platform, e.g. in the footwall of the TFFC (cf. Johansen et al., 1994; Gudlaugsson

et al., 1998) is further explored in the present contribution.

The goal of this paper is to demonstrate the presence of an overall NE-SW trending, NW-dipping, basement-seated, low-angle shear zone on the Finnmark Platform, the Sørøya-Ingøya shear zone (SISZ; Figure 1), and to discuss its role played in shaping the SW Barents Sea margin during late/post-orogenic collapse of the Caledonides in late Paleozoic times and its influence on

the formation and evolution of Devonian-Carboniferous collapse basins. We mapped and analyzed basin-bounding brittle faults on the Finnmark Platform and in the southwesternmost Nordkapp basin, such as the TFFC and the MFC (Figure 1), to evaluate the impact of the SISZ on post-Caledonian brittle faults. We aim at showing the importance of structural inheritance by examining the relationship between Precambrian-Caledonian structural grains, post-Caledonian fault trends

and offshore sedimentary basin geometries. Minor Carboniferous grabens and half-grabens on the Finnmark Platform (e.g. the Sørvær Basin; Figure 1), which are thought to have formed during early stages of extension shortly after the end of the Caledonian Orogeny (Lippard & Roberts,



1987; Olesen et al., 1990; Johansen et al., 1994; Bugge et al., 1995; Gudlaugsson et al., 1998; Roberts et al., 2011), are of particular importance to the present work. We further investigate the

presence of possible Devonian sedimentary deposits on the Finnmark Platform and in the southwesternmost Nordkapp basin and tentatively interpret them as potential analogs to Middle Devonian basins in western Norway (Séranne et al., 1989; Chauvet & Séranne, 1994; Osmundsen & Andresen, 2001) and mid-Norway (Braathen et al., 2000). In this context, NE-SW to ENE-WSW trending basement ridges in the footwall of the TFFC and on the northern flank of the

southwesternmost Nordkapp basin are described and analyzed, and we compare them to adjacent basement highs such as the Norsel High (Figure 1; Gabrielsen et al., 1990; Gudlaugsson et al., 1998), the West Troms Basement Complex (Zwaan, 1995; Bergh et al., 2010) and the Lofoten Ridge (Blystad et al., 1995; Bergh et al., 2007; Hansen et al., 2012). Finally, we propose a model of exhumation of these ENE-WSW to NE-SW trending basement ridges as a metamorphic core

complex (cf. Lister & Davis, 1989) using shear zones in Vesterålen as onshore analogs for the SISZ (Steltenpohl et al., 2004; Osmundsen et al., 2005; Steltenpohl et al., 2011).

## 2. Geological setting

        The bedrock geology of the SW Barents Sea margin (Figure 1) consists of (i) an Archaean

to Paleoproterozoic basement suite, the West Troms Basement Complex (Zwaan, 1995; Bergh et al., 2010), (ii) locally preserved autochthonous Neoproterozoic cover sequences (Kirkland et al., 2008), (iii) a series of Caledonian thrust nappes (Andersen, 1981; Ramsay et al., 1985; Corfu et al., 2014), and (iv) late Paleozoic to Cenozoic sedimentary sequences offshore (Faleide et al., 1993, 2008; Gudlaugsson et al., 1998; Worsley, 2008; Smelror et al., 2009; Figure 1). Archean to

Paleoproterozoic basement rocks are mostly exposed in major horsts and ridges in Western Troms (Bergh et al., 2010; Indrevær et al., 2013; Indrevær & Bergh, 2014), whereas Neoproterozoic and Caledonian rocks dominate in the eastern part of Troms and in NW Finnmark (Kirkland et al., 2008; Corfu et al., 2014; Indrevær & Bergh, 2014; Figure 1). In offshore areas adjacent to Western Troms and NW Finnmark, extensive post-Caledonian normal faulting led to the formation of large

sedimentary basins that are filled with thick, late Paleozoic to Cenozoic deposits related to the post-orogenic collapse of the Caledonides and to the opening of the NE Atlantic Ocean (Faleide et al., 1993, 2008; Gudlaugsson et al., 1998; Worsley, 2008; Smelror et al., 2009). Late Paleozoic-



Cenozoic sedimentary units are missing in onshore areas of Troms and Finnmark likely due to erosion and/or non-deposition (Ramberg et al., 2008; Smelror et al., 2009).


## 2.1. Onshore Precambrian and Caledonian geology

### 2.1.1. Precambrian basement rocks

The Western Troms margin is characterized by Archean to Paleoproterozoic basement
rocks of the West Troms Basement Complex (Bergh et al., 2010) that are preserved and exposed in a horst block formed during post-Caledonian extension (Indrevær et al., 2013). The West Troms Basement Complex consists of tonalitic, trondhjemitic and granitic gneisses, metasupracrustal rocks and mafic and felsic igneous rocks (Corfu et al., 2003; Bergh et al., 2010). These rocks were deformed during the Svecofennian orogeny, which resulted in the formation of NW-SE trending
steep foliation, ductile shear zones and upright and vertical macrofolds, only weakly reworked during the Caledonian Orogeny (Corfu et al., 2003; Bergh et al., 2010).

In NW Finnmark, Paleoproterozoic basement rocks occur in several tectonic windows of the Caledonides, e.g. Repparfjord-Komagfjord and Alta-Kvænangen tectonic windows (Zwaan & Gautier, 1980; Pharaoh et al., 1982; 1983; Bergh & Torske, 1988), and consist of low-grade
supracrustal metavolcanics and metasedimentary rocks of the Raipas Group. These Greenstone belts formed in NW-SE trending rift basins in the Archean?-Paleoproterozoic during the opening of the Kola Ocean (Bergh & Torske, 1986; 1988), although more recent studies tentatively reinterpret these rocks as foreland basin deposits derived from the Svecokarelian Orogeny (Torske & Bergh, 2004). A thin cover of Neoproterozoic to Cambrian (para-) autochtonous
metasedimentary rocks occurs on top of the Paleoproterozoic basement windows. This unit is correlated with the autochthonous Tanafjord-Varangerfjord Group in eastern Finnmark (Siedlecki, 1980), as well as with the Lower Allochthonous Gaissa and Laksefjord Nappes (Ramsay et al., 1985; Corfu et al., 2014).

Another Neoproterozoic unit, the Barents Sea Group (Siedlecki, 1980), is exposed in the
outer part of the Varanger Peninsula (Figure 1). The rocks of the Barents Sea Group were deposited in a large-scale foreland basin during the Timanian Orogeny (Andresen et al., 2014) and a Cryogenian depositional age was inferred from fossil-bearing assemblages (Corfu et al., 2014). The Timanian Orogeny produced major NW-SE trending folds (Roberts & Siedlecka, 2002) and



WNW-ESE trending fault complexes like the TKFZ (Jonhson et al., 1978; Herrevold et al., 2009).

The TKFZ was mapped as a narrow, single-segment fault strand all the way from the Barents shelf in the west (Gabrielsen, 1984; Gabrielsen & Færseth, 1989; Gabrielsen et al., 1990) to the east of the Kola Peninsula in Russia (Roberts et al., 1997). Between these areas, the TKFZ is traced along the Kola Peninsula as the Sredni-Rybachi Fault Zone (Roberts et al., 2011). On the Varanger Peninsula, the TKFZ is well displayed on satellite and DEM images, but is generally poorly

exposed. In map view, the TKFZ is irregular, with different structural segments and branching subsidiary faults both across- and along-strike, locally showing duplex structures (Siedlecka & Siedlecki, 1967; Siedlecka, 1975). The TKFZ formed along the southwestern boundary of the Timanian Orogeny in the late Cryogenian-Ediacaran (Roberts & Siedlecka, 2002; Siedlecka et al., 2004), and was later reactivated as a strike-slip fault during the Caledonian Orogeny when it

accommodated significant lateral displacement constrained to 200-250 km of dextral strike-slip movement (Bylund, 1994; Rice, 2013).

### 2.1.2. Caledonian rocks

Coastal areas of NW Finnmark are dominated by Caledonian thrust sheets of the Kalak

Nappe Complex and Magerøy Nappe (Ramsay et al., 1985; Ramberg et al., 2008; Corfu et al., 2014), formed in the Neoproterozoic through Silurian (Figure 1). The Kalak Nappe Complex is composed of amphibolite facies schists, metapsammites and paragneisses, and comprises several allochthonous thrust sheets with Proterozoic basement rocks, clastic metasedimentary rocks, and plutonic rocks of the Seiland Igneous Province (Corfu et al., 2014). A major thrust defines the

contact with the underlying pre-Caledonian basement (Ramsey et al., 1985). Dominant structures include a gently NW-dipping foliation, NNE-SSW trending, east-verging, asymmetrical recumbent folds and low-angle thrusts that accommodated top-to-the-ESE shortening (Townsend, 1987a; Kirkland et al., 2005). The Kalak Nappe Complex likely corresponds to an exotic terrane that was accreted on the Laurentian margin of Rodinia, prior to the rifting linked to the opening of the

Iapetus Ocean, and to have later been thrusted over Baltica during the Caledonian Orogeny (Kirkland et al., 2008).

The Seiland Igneous Province corresponds to a large, late Neoproterozoic mafic and ultramafic intrusion linked to the early-mid rifting stages of the Iapetus Ocean (Elvevold et al., 1994; Corfu et al., 2014). Recent geophysical studies by Pastore et al. (2016) show that the base of



185 the Seiland Igneous Province defines two deep-reaching roots located below the islands of Seiland and Sørøya constraining the thickness of the Kalak Nappe Complex in this area to a maximum of 10 km. On the Porsanger and Varanger Peninsula, ENE-WSW to NNE-SSW trending, Ediacaran metadolerite dyke swarms are particularly common, and they are as well associated to the rifting of the Iapetus Ocean (cf. Roberts, 1972; Siedlecka et al., 2004; Nasuti et al., 2015).

190   The Kalak Nappe Complex is structurally overlain by the Magerøy Nappe, which consists of Late Ordovician to early Silurian greenschist facies metasedimentary and metaplutonic rocks (Andersen, 1981; 1984; Corfu et al., 2014) that crop out on the island of Magerøya (Figure 1). The Magerøy Nappe is characterized by asymmetrical, NNE-SSW trending, east-verging folds and low-angle, NW- and SE-dipping thrusts similar in trend to those observed within the Kalak Nappe

195 Complex (Andersen, 1981), and is intruded by granitic and gabbroic plutons, e.g. the Silurian Honningsvåg Igneous Complex (Corfu et al., 2006) and the Finnvik Granite (Andersen, 1981). Remnants of the Magerøy Nappe thrust units are also found in northeastern Sørøya and on the Porsanger Peninsula (Kirkland et al., 2005; 2007; Corfu et al., 2014; Figure 1).

   In nearshore areas of NW Finnmark, along the coasts of Sørøya and Ingøya, we identified

200 on seismic sections a large NE-SW trending, dominantly NW-dipping zone of weakness below post-Caledonian basin and faults that we tentatively interpret as a major Caledonian shear zone, which we name the Sørøya-Ingøya shear zone (Figure 1). This large weakness zone has not been described in scientific literature and we therefore proceed by describing its geometry and potential kinematics based on offshore seismic data.


## 2.2. Post-Caledonian brittle faults and basins

### 2.2.1. Post-Caledonian offshore basins

   The SW Barents Sea margin was subjected to multiple episodes of extensional faulting after

210 the end of the Caledonian Orogeny, starting with the collapse of the Caledonides in the Mid/Late Devonian-early Carboniferous and lasting until the early/mid Permian, although evidence of this stage is only preserved onshore western and mid-Norway (Séranne et al., 1989; Chauvet & Séranne, 1994; Braathen et al., 2000; Osmundsen & Andresen, 2001). During this period, basement ridges in Lofoten-Vesterålen (Klein & Steltenpohl, 1999; Klein et al., 1999; Steltenpohl et al.,

215 2004; 2011) and in mid-Norway (Osmundsen et al., 2005) were exhumed as metamorphic core



complexes, synchronously with the development of large half-graben basins such as the Vøring and Møre basins in mid-Norway (Blystad et al., 1995) and the Hammerfest, Nordkapp and Ottar basins in the SW Barents Sea (Gabrielsen et al., 1990; Breivik et al., 1995; Gudlaugsson et al., 1998; Indrevær et al., 2013; Figure 1). The main rifting events occurred in the Late Jurassic and

peaked in the Early Cretaceous, when major offshore basins such as the Tromsø and Harstad basins formed. The rifting ended with full breakup of the North Atlantic Ocean and formation of a transform plate margin in the SW Barents Sea at the Paleocene-Eocene transition (Faleide et al., 1993; 2008).

Off the coasts of Western Troms and NW Finnmark, the SW Barents Sea margin is
characterized by a relatively shallow area, the Finnmark Platform (Gabrielsen et al., 1990; Figure 1), which is thought to have remained relatively stable since late Paleozoic times. For example, the inner part of the Finnmark Platform, here referred to as the Finnmark Platform east (Figure 1), was only affected by the formation of minor Carboniferous, ENE-WSW to NE-SW trending half-graben and graben structures (Bugge et al., 1995; Samuelsberg et al., 2003; Rafaelsen et al., 2008;
Figure 1). In the hanging-wall of the MFC on the western part of the Finnmark Platform, the Finnmark Platform west (Figure 1), shows a prominent gravity low, the Gjesvær Low, which was ascribed to the presence of low-density Caledonian rocks (Johansen et al., 1994; Gernigon et al., 2014). We explore and argue for an alternative explanation, i.e. the presence of Devonian collapse basin deposits draped against a low-angle extensional detachment of the SISZ, similar to the
Nordfjord-Sogn Detachment Zone, a late-orogenic shear zone that bounds the Middle Devonian Hornelen, Kvamshesten and Solund sedimentary basins onshore western Norway (Séranne et al., 1989; Chauvet & Séranne, 1994; Wilks & Cuthbert, 1994; Osmundsen & Andersen, 2001). Ductile detachment surfaces of comparable size, showing analog kinematics and contemporaneous timing of activity as the Nordfjord-Sogn Detachment Zone are documented as far north as the Lofoten-
Vesterålen Margin (Klein & Steltenpohl, 1999; Klein et al., 1999; Steltenpohl et al., 2004; 2011), but Devonian collapse basin sedimentary rocks and extensional detachments have not yet been reported along the margins of Western Troms and NW Finnmark.

### 2.2.2. Post-Caledonian faults and fractures trends

Multiple studies have reported post-Caledonian brittle faults onshore coastal areas in Lofoten-Vesterålen, Western Troms and NW Finnmark (Roberts, 1971; Worthing, 1984; Lippard



& Roberts, 1987; Townsend, 1987a; Rykkelid, 1992; Lippard & Prestvik, 1997; Roberts & Lippard, 2005; Bergh et al., 2007; Hansen et al., 2012; Indrevær et al., 2013; Davids et al., 2013). A common feature is the presence of rhombic, zigzag-shaped fault trends similar in geometry to

offshore basin-bounding faults. Dominant fault-fracture trends of the margin strike NNE-SSW, ENE-WSW and NW-SE, respectively (Bergh et al., 2007; Eig, 2008; Eig & Bergh, 2011; Hansen et al., 2012; Hansen & Bergh, 2012; Indrevær et al., 2013). Typical examples are basin-bounding, NNE-SSW and ENE-WSW trending brittle normal faults that are part of the Vestfjorden-Vanna Fault Complex, which bounds the offshore Vestfjorden Basin southeast of the Lofoten islands and

which can be traced northward to Western Troms (Indrevær et al., 2013; Figure 1), whereas the NW-SE trend typically reflects the margin-oblique, transform fault trends (Faleide et al., 2008). An analog to the onshore Vestfjord-Vanna Fault Complex in NW Finnmark is the Langfjorden-Vargsundet fault (Figure 1), described by Zwaan & Roberts (1978) and Worthing (1984) as a major NE-SW trending, NW-dipping normal fault where rocks from the Kalak Nappe Complex and the

Seiland Igneous Province in the northwest are juxtaposed against Precambrian basement rocks of the Repparfjord-Komagfjord and Alta-Kvænangen tectonic windows in the southeast (Figure 1).

The NW Finnmark margin is located along the northeastward prolongation of the Lofoten-Vesterålen and Western Troms segments of the Norwegian continental shelf (Figure 1). Similar fault sets and trends as in Lofoten-Vesterålen exist in Finnmark and their interaction is thought to

partly have controlled the rhombic geometry of many offshore sedimentary basins (Bergh et al., 2007; Indrevær et al., 2013). A typical example along the Western Troms and NW Finnmark margins is the NW-dipping TFFC, which bounds the Harstad Basin to the east and the Hammerfest Basin to the southeast (Gabrielsen et al., 1990; Indrevær et al., 2013). The TFFC defines a system of irregular branching faults trending NNE-SSW and ENE-WSW and terminating as a WNW-ESE

trending fault zone northwest of the island of Magerøya where it merges with the NE-SW trending, NW-dipping MFC at the southeastern boundary of the Nordkapp Basin (Gabrielsen et al., 1990) and of the triangular-shaped southwesternmost Nordkapp basin (Omosanya et al., 2015; Figure 1). We address a possible genetic relationship and structural inheritance of the post-Caledonian MFC with the Caledonian SISZ and argue that the MFC may have initiated as an extensional splay during

the reactivation of the SISZ as an extensional detachment during the late/post-orogenic collapse of the Caledonides. Furthermore, we tentatively link basement ridges such as the Norsel High in the



footwall of the Nysleppen Fault Complex (Gabrielsen et al., 1990) to bowed segments of the SISZ (Figure 1).

### 280    2.2.3. Post-Caledonian transfer zones

The Norwegian continental shelf is segmented by transfer fault zones of which the largest is the offshore De Geer Zone (Faleide et al., 1984; 2008; Cianfarra & Salvini, 2015), which main fault segment is the Hornsund Fault Zone, an offshore NNW-SSE trending fault that runs parallel to the west coast of Spitsbergen and separates the SW Barents Sea margin from the Lofoten-

Vesterålen Margin (Figure 1). In the south, the De Geer Zone proceeds through the Senja Fracture Zone and into the Senja Shear Belt onshore the island of Senja (Figure 1). Olesen et al. (1993; 1997) suggested shifts of polarity of the Vestfjorden-Vanna Fault Complex along the Senja Fracture Zone, and they argued that the formation of the Senja Fracture Zone offshore was controlled by a major onshore basement weakness zone, the Bothnian-Senja Fault Complex, which

provided suitably oriented basement heterogeneities for the development of a transfer zone (e.g. Doré et al., 1997). Similarly, Indrevær et al. (2013) proposed the existence of a fault array termed the Fugløya transfer zone to explain offsets and shifts of polarity along the Vestfjorden-Vanna Fault Complex farther northeast in WesternTroms. The Fugløya transfer zone trends N-S/NNW-SSE and continues onshore Western Troms, where it merges with the NW-SE trending Bothnian-

Kvænangen Fault Complex, and offshore where it is thought to merge into the TFFC and the Ringvassøy-Loppa Fault Complex (Indrevær et al., 2013; Figure 1).

Analogously in NW Finnmark, the WNW-ESE trending TKFZ seems to merge into a basin-bounding fault, in this case the WNW-ESE trending, NE-dipping fault segment of the TFFC (Gabrielsen, 1984; Gabrielsen & Færseth, 1989; Roberts et al., 2011). In nearshore areas of NW

Finnmark, the TKFZ is thought to proceed offshore and seems to correlate with a large escarpment north of Magerøya and into the Barents Sea (Vorren et al., 1986; Townsend, 1987b). In the area where it terminates, it merges and links up with the TFFC to form triangular-shaped mini-basins (Gabrielsen, 1984; Gabrielsen & Færseth, 1989; Roberts et al., 2011). We explore an alternative origin for the WNW-ESE trending fault segment of the TFFC and further examine its interaction

with the onshore-nearshore TKFZ, which potentially acted as a transfer fault after the Caledonian Orogeny and contributed to offset the LVF onshore Magerøya and in adjacent coastal areas (Koehl et al., submitted; Figure 1). Other major WNW-ESE trending faults exist offshore, northeast of the

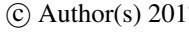



Varanger Peninsula, and these bound the Tiddlybanken Basin, a large WNW-ESE trending basin that formed in Carboniferous times (Mattingsdal et al., 2015; Figure 1).


### 2.2.4. Absolute age dating of post-Caledonian faults

The absolute age of post-Caledonian brittle faults onshore NW Finnmark is poorly constrained although a few contributions provide valid insights (Lippard & Prestvik, 1997; Davids et al., 2013; Torgersen et al., 2014; Koehl et al., 2016). Torgersen et al. (2014) performed K/Ar

dating of brittle fault gouge in the footwall of the LVF and obtained dominantly Carboniferous to early Permian ages, as well as a subsidiary Early Cretaceous age for one of the faults. Roberts et al. (1991) and Lippard & Prestvik (1997) presented indirect evidence of early Carboniferous dolerite dykes emplaced along and cementing WNW-ESE trending brittle fault segments of the TKFZ onshore Magerøya. These dykes produce high positive aeromagnetic anomalies (Nasuti et

al., 2015) and may be used to further identify brittle faults in NW Finnmark. Late Devonian dolerite dykes emplaced along brittle faults that trend NE-SW and N-S have been identified and dated in eastern Varanger Peninsula (Guise & Roberts, 2002) and on the Kola Peninsula (Roberts & Onstott, 1995). By comparison, Davids et al. (2013) obtained Late Devonian-early Carboniferous ages from K/Ar dating of illite clay minerals for early extensional faulting along the Vestfjorden-Vanna Fault

Complex and related faults in Lofoten-Vesterålen and Western Troms.

### 2.3.    Offshore sedimentary successions and well-ties

Deep fault-bounded basins formed along the SW Barents Sea margin during successive

extension events in late Paleozoic-early Cenozoic times, and  these basins contain important sedimentary successions for hydrocarbon exploration. We particularly focus on the late Paleozoic succession which sedimentary rocks were deposited on top of eroded Precambrian and Caledonian basement rocks (cf. Townsend, 1987a; Johansen et al., 1994; Bugge et al., 1995; Zwaan, 1995; Gudlaugsson et al., 1998; Samuelsberg et al., 2003; Bergh et al., 2010). Late Paleozoic sedimentary

deposits in the study area were penetrated by only a few exploration wells to which we tied our seismic interpretation. These wells include exploration wells 7120/12-4, 7124/3-1, 7128/4-1 and 7128/6-1, and shallow drill-cores 7127/10-U-2 and 7127/10-U-3 from Bugge et al. (1995; Figure



2). Overlying Mesozoic to Cenozoic sedimentary units were not investigated and are better described in Omosanya et al. (2015).

The nature and age of basement rocks along the SW Barents Sea margin remain relatively complex to resolve since only a handful of wells drilled through the thick post-Caledonian sedimentary cover. Nevertheless, wells 7128/4-1 and 7128/6-1 penetrated quartzitic metasedimentary rocks on the Finnmark Platform east (Figure 2) and these are believed to correlate with upper Proterozoic rocks involved in Caledonian thrusting in northern Finnmark (Røe &

Roberts, 1992).

      Devonian sedimentary rocks are yet to be reported in North Norway and along the SW Barents Sea margin. Devonian sedimentary deposits however are present in western Norway (Osmundsen & Andersen, 2001) where they represent a several km-thick succession made up with clastic deposits that notably include rhythmic sandstone and coarse-grained conglomerate units that

were deposited in the hanging-wall of a major, low-angle extensional shear zone, the Nordfjord-Sogn Detachment Zone (Séranne et al., 1989; Wilks & Cuthbert, 1994; Osmundsen & Andersen, 2001).

      Lower Carboniferous sedimentary rocks of the Billefjorden Group directly overlie basement rocks on the Finnmark Platform east as evidenced by exploration wells 7128/4-1 and

7128/6-1 (Larssen et al., 2002; Figure 2). These rocks mostly correspond to fluvial clastic deposits interbeded with coal-bearing sedimentary rocks that correlate with contemporaneous deposits onshore Bjørnøya (Cutbill & Challinor, 1965; Gjelberg, 1981, 1984) and Spitsbergen (Cutbill & Challinor, 1965; Cutbill et al., 1976; Gjelberg, 1984). The total thickness of Billefjorden Group sedimentary deposits evidenced by exploration wells on horst-blocks on the Finnmark Platform

east ranges from 350 m to 450 m. However, in the hanging-wall of a minor normal fault interpreted by Bugge et al. (1995) near the coast of northern Finnmark (Figure 2), shallow drill-cores 7127/10-U-2 and 7127/10-U-3 indicate that the thickness of lower Carboniferous sedimentary rocks reaches a thickness > 600 m within a NE-SW trending mini-basin on the Finnmark Platform east near the coast of the Nordkinn Peninsula (cf. star symbol in Figure 1 & Figure 2). In the Serpukhovian,

fluvial sediments of the Billefjorden Group were gradually replaced by shallow marine sediments of the Gipsdalen Group from which they are generally separated by a mid-Carboniferous (Serpukhovian) unconformity (Cutbill et al., 1976; Gjelberg, 1984) potentially related to a global



sea-level fall (Saunders & Ramsbottom, 1986). This unconformity was recognized on the Finnmark Platform east by Bugge et al. (1995).

Shallow marine sedimentary deposits of the Gipsdalen Group are widespread along the SW Barents Sea margin and have proven prolific for hydrocarbon exploration (Larssen et al., 2002). Thus, this sedimentary succession benefits from a relatively high number of well penetrations and as a result, its facies and lateral thickness variations are well-constrained (Samuelsberg et al., 2003; Rafaelsen et al., 2008). The Gipsdalen Group was notably penetrated by wells 7128/4-1 and

7128/6-1 on the Finnmark Platform east, by well 7120/12-4 on the Finnmark Platform west and by well 7124/3-1 on the northern flank of southwesternmost Nordkapp basin (Larssen et al., 2002; Figure 2). Sedimentary strata of the Gipsdalen Group were tied to their stratigraphic equivalent onshore Bjørnøya and Spitsbergen (Gjelberg & Steel, 1981, 1983; Gjelberg, 1984; McCann & Dallmann, 1996). This succession consists of alluvial clastic sedimentary rocks deposited on top

of lower Carboniferous sedimentary deposits of the Billefjorden Group (McCann & Dallmann, 1996). In the upper part of the Gipsdalen Group, alluvial clastic sedimentary rocks are progressively replaced by shallow marine platform carbonates interbedded with clastic and evaporite deposits. In well 7124/3-1 (Figure 2), Asselian evaporite deposits typically include thin layers of anhydrite and gypsum, but thicker, halite-rich end-members are found along the flanks of

the Nordkapp Basin and southwesternmost Nordkapp basin where large pillows of upper Carboniferous and lower Permian salt were observed (Gabrielsen et al., 1992; Jensen & Sørensen, 1992; Koyi et al., 1993; Nilsen et al., 1995; Gudlaugsson et al., 1998; Koehl et al., 2017). In the Nordkapp Basin, pre-Permian deposits may in places reach a thickness of up to 7-8 km (Gudlaugsson et al., 1998). These deposits are composed of thick clastic sedimentary rocks and of

upper Carboniferous to lower Permian evaporite deposits characterized by mobile salt that was involved in salt tectonism in the southwesternmost Nordkapp basin (Gudlaugsson et al., 1998; Koehl et al. 2017) and in the Nordkapp Basin (Gabrielsen et al., 1992; Jensen & Sørensen, 1992; Koyi et al., 1993; Nilsen et al., 1995).

        In the present work, we renamed the sedimentary basin located at the intersection of the

TFFC and MFC (Figure 1) as the "southwesternmost Nordkapp basin" instead of the "easternmost Hammerfest basin" as proposed by Omosanya et al. (2015). The reason is that this basin shows more geometric similarity to the Nordkapp Basin than the Hammerfest Basin and must be treated as a separate basin (see further arguments later).



## 3. Methods & databases

### 3.1. Seismic data and well-ties

The seismic interpretation shown in this study is based on publicly available 2D and 3D data from the DISKOS database. The interpretation of seismic data aims at providing good constraints for the extent and geometry of offshore brittle faults and for offshore stratigraphy on the Finnmark Platform and in the southwesternmost Nordkapp basin. The present study uses ties to wells 7120/12-4, 7128/4-1 and 7128/6-1 and 7124/3-1. Seven seismic profiles from the BSS01 2D seismic survey were used to analyze and describe offshore basin and fault geometries and provide the basis for discussion about the late Paleozoic evolution of the SW Barents Sea margin. Note that none of the seismic profiles used was depth-converted. Therefore, all relevant estimates of fault offsets and stratigraphic seismic unit thicknesses will be described in seconds (s) two-way time (TWT).

### 3.2. Aeromagnetic anomaly data

The offshore aeromagnetic data used in this study correspond to a compilation of the BASAR project of the Geological Survey of Norway (NGU) published by Gernigon & Brönner (2012) and Gernigon et al. (2014; Figure 3). The dataset is composed of tilt derivatives of aeromagnetic data and has been used to delineate possible magmatic intrusions (dykes) emplaced along brittle faults (cf. Nasuti et al., 2015) and abrupt changes of lithology generally recorded across major faults, thus, contributing to the mapping of post-Caledonian offshore brittle faults along the SW Barents Sea margin. However, data uncertainties arise from the fact that significantly different rock types may yield similar aeromagnetic responses. A crucial example in northern Finnmark is the similar high positive narrow aeromagnetic anomalies produced both by sub-vertical folded beds of metasedimentary rocks (Roberts & Siedlecka, 2012; Roberts & Williams, 2013) and dolerite dykes intruded along brittle faults (Nasuti et al., 2015; Figure 3). In order to distinguish such features, we carefully analyzed onshore geology in coastal areas of NW Finnmark and the results of exploration wells on the Finnmark Platform and adjacent offshore basins.




## 4. Results

### 4.1. Seismic interpretation of offshore basins and faults

#### 4.1.1. Seismic units and stratigraphy

On seismic data (Figure 4), basement rocks typically show chaotic internal reflection patterns, which complicate the task of identifying intra-basement structures and basins, and individualize layered sedimentary sequences. However, km-thick layers bearing strong basement fabrics such as widespread gently dipping foliation or pronounced mylonitic fabric commonly

found along large shear zones may turn out to be resolvable at seismic scale (see chapter 4.1.2.). For instance, we observed a several km-thick, curved, shallow-dipping layer that is characterized by moderate-amplitude reflections, which are parallel to the layer's upper and lower boundaries (see "Sørøya-Ingøya shear zone" reflections in Figure 4c-g). We interpret these pronounced internal fabrics as widespread mylonitic foliation within a large-scale shear zone. Numerous

smaller basement shear zones may be present below late Paleozoic-Cenozoic sedimentary rocks on the Finnmark Platform west and these correspond to steeply to moderately dipping fabrics made of sub-parallel, moderate- to high-amplitude reflections (cf. Figure 4b, e, f, g & Figure 5a-c).

Potential Devonian sedimentary deposits along the SW Barents Sea are sparse and as a result their seismic character is not well constrained. This sedimentary succession has not been

drilled, which makes its interpretation on seismic data rather speculative. However, we believe that the best two candidates to represent Devonian sedimentary deposits analog to those in western and mid-Norway (Braathen et al., 2000; Osmundsen & Andersen, 2001) are located at the base of the southwesternmost Nordkapp basin and on the Finnmark Platform west near the Gjesvær Low (Figure 1). In the southwesternmost Nordkapp basin, possible Devonian sedimentary strata are

located at a deep level (below 4 seconds TWT) and their seismic signature is thus largely masked by overlying sedimentary successions (Figure 4c & d). By contrast, on the Finnmark Platform west (Figure 4e), potential Devonian sedimentary rocks are relatively shallower, which makes their seismic pattern easier to distinguish from underlying basement rocks, and from overlying Carboniferous sedimentary deposits and seismic artifacts (Figure 4e). Devonian sedimentary rocks

on the Finnmark Platform west display relatively low seismic amplitudes. The internal reflection pattern is rather chaotic apart from a few discrete, shallow-dipping, moderate-amplitude reflections





that converge towards each other upwards and that we interpret as major sedimentary sequence boundaries (cf. "Base Devonian" reflection in Figure 4e & Figure 5b & c). Furthermore, Devonian sedimentary deposits are likely separated from underlying basement rocks by an angular

unconformity that appears as arcuate, high-amplitude seismic reflections ("Base Devonian" reflection in Figure 4e). We interpret these arcuate, high-amplitude seismic reflections as an erosional unconformity.

Lower Carboniferous sedimentary deposits of the Billefjorden Group, composed of thick clastic sedimentary deposits interbedded with ocasional coal-bearing sedimentary rocks, may

produce high-amplitude seismic reflections related to their organic-rich content (Figure 4a & b). Such sedimentary strata are present on the Finnmark Platform east, where they appear to thicken to the southeast near the coasts of NW Finnmark (Figure 5d), whereas they are rather sparse on the Finnmark Platform west, i.e. eroded or never deposited (Figure 4e & f). On the Finnmark Platform east, the transition from basement rocks (cf. "Top basement" reflection in Figure 4a & b) to lower

Carboniferous sedimentary rocks is difficult to interpret on seismic sections. This is attributable to the strong similarities between high seismic amplitudes displayed locally both by basement rock fabrics such as major shear zones (cf. yellow dotted lines in Figure 4b) and by lower Carboniferous coal-bearing sedimentary deposits. Low amplitude reflections also show identical chaotic patterns in both basement rocks and clastic sedimentary rocks of the Billefjorden Group (Figure 4a & b).

In the southwesternmost Nordkapp basin, lower Carboniferous sedimentary strata are believed to be present although their seismic signature certainly appears to be affected by overlying upper Carboniferous evaporite deposits (Figure 4c & d). The boundary between lower Carboniferous sedimentary deposits and potential underlying Devonian sedimentary rocks was not identified in the southwesternmost Nordkapp basin. Nevertheless, since the maximum thickness of Billefjorden

Group sedimentary strata is ca. 600 m on the Finnmark Platform east (Bugge et al., 1995), this suggests that the several km-thick succession below the mid-Carboniferous reflection and above a thick shear zone in the southwesternmost Nordkapp basin is composed of lower Carboniferous sedimentary rocks probably complemented by thick Devonian sedimentary deposits (Figure 4c & d). Alternatively, sedimentary deposits of the Billefjorden Group directly overlie basement rocks.

On the Finnmark Platform (Figure 1 & Figure 2), the base of upper Carboniferous sedimentary sequences is difficult to identify (cf. "mid-Carboniferous" reflection in Figure 4) because it often appears as a linear, moderate to low amplitude seismic reflection that separates



subparallel reflections of lower and upper Carboniferous sedimentary rocks, whereas in other places this reflection is irregular and truncates either high-amplitude coal-bearing sedimentary
deposits of the Billefjorden Group, high-amplitude reflections produced by basement rocks (Figure 5a) similar to those of the km-thick shear zone below the Finnmark Platform west and the southwesternmost Nordkapp basin (see Figure 4c-g), and/or low-amplitude reflections in Devonian sedimentary strata (Figure 5b & c). Nevertheless, this reflection generally corresponds to an angular unconformity (e.g. Figure 5a-c & e) and is therefore interpreted to correspond to a regional
erosion surface.

In the southwesternmost Nordkapp basin, the base of upper Carboniferous sedimentary deposits (cf. "mid-Carboniferous" reflection in Figure 4c & d) is relatively easy to interpret as it mostly appears as a discrete high-amplitude reflection. The strong acoustic impedance contrast producing the high seismic amplitude for the mid-Carboniferous reflection most likely arises from
the presence of upper Carboniferous evaporite deposits partly composed of mobile salt (halite), which is significantly less dense than regular sedimentary rocks, (cf. "Top upper Carboniferous evaporites" reflection in Figure 4c & d). This evaporite succession was identified by Gudlaugsson et al. (1998) and is restricted to basinal areas located northwest of the MFC and north of the TFFC (Figure 1 & Figure 2). It is characterized by a highly variable thickness, which is due to the presence
of lensoidal bodies bounded to the top and bottom by high-amplitude reflections on the basin edges and to the occurrence of thick bodies made of chaotic reflection patterns near the center of the basin (Figure 4c). We interpret the lensoidal bodies on the basin edges as pillows of mobile salt and the chaotic bodies near the basin center as small salt diapirs based on similarities with large salt diapirs and evaporite deposits observed in the Nordkapp Basin (Gabrielsen et al., 1992; Jensen &
Sørensen, 1992; Koyi et al., 1993; Nilsen et al., 1995). We consider that the presence of analog late Paleozoic evaporite deposits in the southwesternmost Nordkapp basin and in the Nordkapp Basin (Jensen & Sørensen, 1992; Koyi et al., 1993; Gudlaugsson et al., 1998) and the absence of such deposits in the Hammerfest Basin constitute strong enough arguments to justify the proposed change of name for the "easternmost Hammerfest basin" (Omosanya et al., 2015) into the
"southwesternmost Nordkapp basin". Nonetheless, the southwesternmost Nordkapp basin shows large amount of normal displacement along its southern boundary fault, the NW-dipping MFC, which is opposite to the Nordkapp Basin where basin subsidence was dominantly accommodated along the SE-dipping Nysleppen Fault Complex (Figure 1). Hence, despite their similarities, the



Nordkapp Basin and the southwesternmost Nordkapp basin should be treated as two separate
basins.

Non-evaporitic upper Carboniferous and Permian sedimentary deposits are characterized
by subparallel, flat-lying to shallow-dipping, homogeneous, moderate to low-amplitude seismic
reflections (see Figure 4). Permian deposits are relatively thin on the Finnmark Platform and are
sometimes difficult to distinguish from upper Carboniferous deposits (Figure 4a, b, e, f & g). In
the southwesternmost Nordkapp basin, however, late Paleozoic sedimentary deposits are
reasonably thick and individual units are therefore easier to identify at seismic scale and we
interpreted a thin unit characterized by high-amplitude reflections (cf. "Base Asselian" and "Top
Asselian evaporites" reflections in Figure 4c & d) as Asselian (earliest Permian) evaporite deposits
that were evidenced by exploration well 7124/3-1 on the northern flank of the southwesternmost
Nordkapp basin (Figure 2, Figure 4c & d). Where present, this thin Asselian evaporite succession
defines the base of the Permian sedimentary succession and therefore serves as an upper boundary
for the Carboniferous succession (cf. "Base Asselian" reflection in Figure 4c & d). However,
Asselian evaporites are too thin and too discontinuous to be seismically resolvable on the Finnmark
Platform (Bugge et al., 1995). Occasionally, Asselian evaporites are truncated by chaotic
reflections of small salt diapirs sourced from deeper upper Carboniferous evaporites in the
southwesternmost Nordkapp basin (Figure 4c).

The Base Triassic reflection (see Figure 4) defines the (near-) top of the late Paleozoic
sedimentary succession and is easily interpreted through the whole Barents Sea as it corresponds
to a high-amplitude reflection that represents the top of a regionally widespread carbonate unit
(Bugge et al., 1995). Other important seismic reflections interpreted in the present study include
the Base Cretaceous, Base Paleocene, the Upper Regional Unconformity (URU), which
corresponds to a major erosional unconformity and represents the base of Quaternary sediment
cover (Solheim & Kristoffersen, 1984), and the seabed reflection (Figure 4). These reflections are
penetrated by a large number of exploration wells and shallow drill-cores both on the Finnmark
Platform and in the southwesternmost Nordkapp basin, where they all display consistently high
seismic amplitudes (Faleide et al., 1984; Bugge et al., 1995; Gudlaugsson et al., 1998; Omosanya
et al., 2015).



### 4.1.2. Structural architecture of the Finnmark Platform and of the southwesternmost

**Nordkapp basin**

In this section, we describe the most important structural elements of the Finnmark Platform and of the southwesternmost Nordkapp basin (cf. Figure 1 & Figure 2) based on interpreted key seismic sections (Figure 4). We also highlight the most dominant fault trends and their interaction with major structures such as the TFFC, MFC, TKFZ and SISZ to form offshore sedimentary

basins.

*Faults and shear zones within basement rocks*

We identified a several km-thick, curved, shallow-dipping layer of moderate-amplitude reflections that we interpreted to represent a large-scale basement-seated shear zone, which we

name the SISZ. The upper boundary surface of the SISZ (Figure 6) appears to be relatively shallow in coastal areas on the Finnmark Platform west. On the Finnmark Platform west, the SISZ dominantly dips to the NW but switches to a dominant dip to the northeast on the Finnmark Platform east. In the footwall of the MFC and in the southwestern part of the Finnmark Platform west, the SISZ occurs at relatively shallow depth (< 1.5 s TWT) and is believed to have been deeply

eroded and is now overlain by a very thin sedimentary cover (cf. Figure 4c-f & Figure 5d). The SISZ shows significant lateral thickness variations that range from 2.0-2.5 seconds (TWT) near the coastline and in the footwall of the TFFC to 0.5 second (TWT) below the MFC and the TFFC (Figure 4f). The SISZ deepens to the northwest towards the center of the Finnmark Platform west before bending upwards in the footwall of the TFFC (Figure 4e & f). The SISZ then curves down

and merges with the listric TFFC at depth, thus delineating an elongated, NE-SW trending ridge in the footwall of the TFFC (cf. "basement highs" in Figure 1 and Figure 4e &f). A similar pattern is observed in the southwesternmost Nordkapp basin where the SISZ deepens to the northwest before curving up near the center of the basin and merging with the N-boundary fault of the southwesternmost Nordkapp basin, the Rolvsøya fault, hence giving this basin a characteristic "U"

shape in cross-section (Figure 4c & d). The SISZ also curves down in the footwall of the Rolsøya fault and defines a second elongated, ENE-WSW trending ridge (cf. "basement highs" in Figure 1). Importantly, the two basement ridges located in the footwall of the TFFC and of the Rolvsøya fault ("basement highs" in Figure 1), respectively, are separated by a narrow trough that is bounded to the southwest by the WNW-ESE trending segment of the TFFC (Figure 6). Apart from this





narrow trough, the attitude of the SISZ is uniform along NE-SW transects on the Finnmark
        Platform west and within the southwesternmost Nordkapp basin with a gentle dip to the northeast
        (Figure 4g).

                Noteworthy, the spoon-shaped geometry of the SISZ, with asymmetric, NE-SW trend,
        northeastwards-broadening, NE-plunge (Figure 6) appears to coincide with a basement gravity low
on the Finnmark Platform west: the Gjesvær Low (Johansen et al., 1994; Gernigon et al., 2014;
        Figure 1) The geometry of the SISZ also matches the trend and shape of the southwesternmost
        Nordkapp basin (Figure 1 & Figure 6). Farther south, along the coasts of Western Troms and
        westwards below the Hammerfest Basin, the low quality of available seismic data did not allow us
        to trace the SISZ more precisely (Figure 6). On the Finnmark Platform east, the SISZ bends from
NE-SW into a more WNW-ESE trend and changes dip from gentle to steep to the northeast (Figure
        6), and as a result, the SISZ becomes too deep to interpret on seismic data in the northeastern part
        of the Finnmark Platform east (Figure 6). The multiple changes of trend, dip direction, dip angle
        and thickness of the SISZ gives the shear zone a spoon-shaped geometry (Figure 6).

                On the Finnmark Platform west, subsidiary steep, SE-dipping high-amplitude reflections
occur in basement rocks and these are truncated by the mid-Carboniferous reflection and the Base
        Devonian erosional unconformity in the footwall of the TFFC (see yellow dotted lines in Figure
        4e-g). Despite dipping southeast, these reflections resemble the dominant reflection pattern
        observed within the SISZ (Figure 4e & f). Thus, we interpret them as SE-dipping mylonitic shear
        zones (yellow dotted lines in Figure 4e-g). The upper boundary of one of these SE-dipping shear
zones coincides with an abrupt seismic facies change on the Finnmark Platform west, from
        moderately dipping in the west to gently dipping/sub-horizontal low-amplitude seismic reflections
        in the east (Figure 4g). This change also coincides with a ca. one second (TWT) deepening of the
        upper boundary of the SISZ towards the northeast (Figure 4g), and with a small normal offset of a
        lensoidal, eastwards-thickening layer of sub-horizontal reflections located above the SISZ (cf.
dotted black lines in Figure 4g). We interpret these changing attributes to be related to the presence
        of a NNE-SSW trending, ESE-dipping brittle fault that flattens and soles into the SISZ and which
        may have developed along a pre-existing, steep ductile shear zone (yellow dotted lines in Figure
        4g). This fault is likely early Carboniferous in age since it is truncated by the "mid-Carboniferous"
        reflection and does not propagate through overlying late Paleozoic-Cenozoic sedimentary rocks
(Figure 4g).



Similar NE-SW trending but NW-dipping shear zones may exist in basement rocks on the Finnmark Platform east, for example in the form of steeply dipping, high-amplitude seismic reflections truncated by the mid-Carboniferous reflection (cf. yellow dotted lines in Figure 4b and Figure 5a). These reflections differ from gently dipping, high-amplitude reflections of lower

Carboniferous coal-bearing sedimentary deposits (Figure 5a) and rather resemble the SISZ reflection pattern, though these are located well above the presumed continuation of the SISZ (Figure 4e & f). We therefore interpret these steep reflections as a NE-SW trending, NW-dipping shear zone similar to the SISZ (Figure 4b).

*Faults within late Paleozoic sedimentary successions*

Faults that structured the Paleozoic sedimentary strata and basins include the major TFFC and MFC and numerous faults on the Finnmark Platform. The TFFC is made of alternating ENE-WSW and NNE-SSW trending, NW-dipping, listric fault segments that form a zigzag pattern and that separate the Hammerfest Basin in the northwest from the Finnmark Platform west in the

southeast (Figure 1 & Figure 4e & f; Gabrielsen et al., 1990; Indrevær et al., 2013). Seismic data below ENE-WSW and NNE-SSW trending fault segments of the TFFC show that these fault segments merge with and sole into shallow-dipping reflections of the SISZ at depth (Figure 4e & f). At the northeast termination of the Hammerfest Basin, the TFFC bends 90 degrees clockwise and continues to the southeast as a WNW-ESE trending, NE-dipping, listric fault (Figure 1, Figure

2 & Figure 4g). At depth, this fault merges with the SISZ (cf. Figure 4g) near a narrow trough in the top surface of the SISZ, separating two elongated NE-SW to ENE-WSW trending basement ridges in the footwall of the TFFC and of the Rolvsøya fault (cf. "basement highs" in red in Figure 1 & Figure 6). In map-view, the WNW-ESE trending, NE-dipping fault segment of the TFFC bends anticlockwise into the main fault segment of the MFC, which corresponds to a linear, NE-SW

trending, NW-dipping fault (Figure 1, Figure 2 and Figure 7a & b). The interaction of these two faults in map-view gives the Finnmark Platform west and the southwesternmost Nordkapp basin triangular shapes (Figure 2 & Figure 7a & b). The main fault segment of the MFC defines the southeastern boundary of the southwesternmost Nordkapp basin (Figure 1, Figure 2 & Figure 4c & d) and of a ca. 25-30 km wide graben structure on the Finnmark Platform west that is believed

to be partly filled with Devonian sedimentary deposits (Figure 1, Figure 2 & Figure 4e & f). Northeastwards, the main segment of the MFC (Figure 4c-f) is replaced by several minor fault

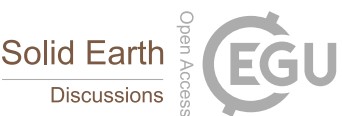

segments of the MFC with limited vertical throw (Figure 4a & b) that defines the southeastern boundary of the Nordkapp Basin (Figure 1 and Figure 4a & b). The southwesternmost Nordkapp basin is bounded to the north by an E-W to ENE-WSW trending, south-dipping, listric normal fault,

the Rolvsøya fault, which flattens at depth and merges into gently dipping reflections of the SISZ (Figure 4c & d). The Rolvsøya fault separates the southwesternmost Nordkapp basin from the Ottar Basin to the northwest and from the Nordkapp Basin to the northeast (Figure 1 & Figure 2).

Late Paleozoic grabens on the Finnmark Platform east display fault patterns that are analogous to those that shape the southwesternmost Nordkapp basin and the Finnmark Platform

west (Figure 1 & Figure 2). Numerous steeply dipping, listric normal faults made of alternating, zigzag-shaped, ENE-WSW and NNE-SSW trending fault segments bound relatively narrow, few km-wide graben and half-graben structures that are filled with wedge-shaped, late Paleozoic sedimentary sequences (Figure 2 & Figure 4a & b). In particular, one of these zigzag-shaped faults trends NE-SW to NNE-SSW, dips to the northwest and can be traced for about 60 km from the

northern coast of Magerøya onto the Finnmark Platform east (Figure 1 & Figure 2). Southwestwards, this fault roughly aligns with a similarly shaped and oriented, NW-dipping onshore-nearshore fault complex synthetic to the TFFC described as the LVF (Figure 2 & Figure 4a & b; Zwaan & Roberts, 1978; Lippard & Roberts, 1987; Roberts & Lippard, 2005; Koehl et al., submitted). We tentatively interpret the ca. 60 km-long, zigzag-shaped brittle fault on the Finnmark

Platform east, northeast of Magerøya, as the northeastward continuation of the LVF on the Finnmark Platform east (Figure 1, Figure 4a & b and Figure 5a).

Below the minor northern fault segments of the MFC, we identified a large NE-SW trending, SE-dipping fault that is antithetic to the MFC (Figure 4a & b). Due to the rather low quality of seismic data at large depths, the interaction of the northern fault segments of the MFC

with the antithetic SE-dipping fault is difficult to evaluate. Our data indicate that the northern fault segments of the MFC crosscuts the NE-SW trending, SE-dipping fault in the southwest (Figure 4b), whereas farther northeast along strike, the northern fault segments of the MFC seem to sole into upper Carboniferous evaporite deposits (Figure 4a).

**4.1.3. Fault-controlled thickness variations**

In the following chapter, fault offsets and thickness variations in the sedimentary successions across brittle faults will be described as a basis to infer timing and sense of shear for



brittle faults on the Finnmark Platform and in the southwesternmost Nordkapp basin. Regional stratigraphic thickness maps (Figure 8a-c) show that late Paleozoic sedimentary strata on the Finnmark Platform east thicken from < 0.1 second (TWT) in the southeast to a maximum thickness of ca. 2 seconds (TWT) in the footwall of the MFC (see also Figure 4a & b). This gradual thickness increase contrasts with the abrupt thickness increase of Devonian-Carboniferous sedimentary strata in the hanging-wall of major normal faults, e.g. the WNW-ESE trending segment of the TFFC and the main segment of the MFC (Figure 8a-b), thus separating depositional versus tectonic thickness changes.

*Intra-basement thickness changes*

The dominant shear zone system within basement rocks on the Finnmark Platform west is the SISZ (Figure 4c-g, Figure 5b-c & e-f and Figure 6). A pronounced intra-basement unit made of sub-horizontal, high-amplitude reflections occurs above the SISZ (Figure 4g).The top reflection of the SISZ and the overlying intra-basement unit are offset by a NNE-SSW trending, gently east-dipping fault, which is accompanied by thickness increase of the intra-basement unit across the east-dipping fault (cf. black dotted line in Figure 4g & Figure 5e). This fault is interpreted to have a top-to-the-E, normal sense of shear (cf. dotted black lines in Figure 4g & Figure 5e), and is itself truncated by the subhorizontal mid-Carboniferous reflection, which constrains its activity to the Mid/Late Devonian-early Carboniferous (Figure 4g).

*Fault-controlled thickness changes in Devonian-Carboniferous strata*

In the southwesternmost Nordkapp basin, the Devonian-lower Carboniferous sedimentary succession (Figure 4c & d) appears to be thickest at the intersection of the TFFC and MFC (Figure 8a), where vertical displacement along the MFC and TFFC is estimated to be ca. 1.5 second (TWT), based on offset of the mid-Carboniferous reflection (cf. Figure 4d). The overlying upper Carboniferous succession displays a similar attitude as shown by the broad thickening of similar sedimentary strata at the intersection of the TFFC and MFC (Figure 8b). These observations suggest that the WNW-ESE trending segment of the TFFC and the main segment of the MFC potentially formed simultaneously in Devonian times and acted as syn-sedimentary normal faults that contributed to the thickening of Devonian-lower Carboniferous and upper Carboniferous sedimentary deposits within the southwesternmost Nordkapp basin (Figure 4c & d). In this



scenario, the Rolvsøya fault likely limits the extent of thickened Devonian-lower Carboniferous and upper Carboniferous sedimentary strata to the north. If we consider the thickness of the seismic package limited upwards by the mid Carboniferous reflection and downwards by the top reflection of the SISZ in the footwall of the Rolvsøya fault, the maximum thickness of Devonian and lower Carboniferous sedimentary rocks on the northern flank of the basin does not exceed ca. 1 second (TWT). This thickness estimate is significantly thinner than what is observed within the

southwesternmost Nordkapp basin, where the Devonian-lower Carboniferous succession reaches a maximum thickness of ca. 2-2.5 seconds (TWT; see Figure 4c & d). By analogy, the thickness of upper Carboniferous sedimentary strata on the northern flank of the southwesternmost Nordkapp basin decreases from ca. 1.5 seconds (TWT) to ca. 0.5-1 second across the Rolvsøya fault (Figure 4c & d and Figure 8b). Hence, the Rolvsøya fault was active and largely contributed to sediment

thickening within the southwesternmost Nordkapp basin during the Mid/Late Devonian-Carboniferous.

On the Finnmark Platform west, potential Devonian sedimentary rocks are characterized by low-amplitude, chaotic reflections within which we observed distinct, shallow-dipping, moderate-amplitude reflections that we interpreted as major sedimentary sequence boundaries (cf.

white dotted lines in Figure 4e and Figure 5b & c). These shallow-dipping reflections diverge from each other downwards and define gently dipping, wedge-shaped layers of low-amplitude, chaotic reflections that thicken downwards against arcuate, high-amplitude basement reflections that represent an erosional unconformity (cf. "Base Devonian" reflection in Figure 4e), and to the northwest against an ENE-WSW trending, SE-dipping normal fault (Figure 4e and Figure 5b & c).

We interpret these sedimentary sequences separated by shallow-dipping, moderate-amplitude reflections to represent growth strata deposited along an active ENE-WSW trending, SE-dipping normal fault, which is parallel to SE-dipping basement shear zones (Figure 4e and Figure 5b & c). In addition, the main fault segment of the MFC shows decreasing amount of vertical displacement to the southwest accompanied by a simultaneous thickness decrease in the upper Carboniferous

succession along the main segment of the MFC (Figure 8b) before the MFC eventually dies out on the Finnmark Platform west (Figure 1, Figure 2 & Figure 4e & f). Analogously, upper Carboniferous sedimentary deposits on the Finnmark Platform west display a wedge shape that is thickest near the MFC and gradually thins towards the TFFC in the northwest (Figure 4e & f, Figure

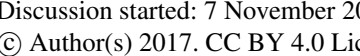


8b). This upper Carboniferous sedimentary wedge likely formed by syn-tectonic sedimentary

growth along the main fault segment of the MFC.

On the Finnmark Platform east, the offshore portion of the LVF (cf. Figure 1 & Figure 4a & b) downthrows the mid-Carboniferous reflection by ca. 0.5 second (TWT) to the northwest (Figure 4b) and bounds a NE-SW trending graben structure filled with thickened lower Carboniferous and upper Carboniferous sedimentary strata (cf. Figure 4a & b). In this graben

structure, the lower Carboniferous and upper Carboniferous sedimentary successions thicken against the LVF (Figure 4b), while thickness variations become negligible farther north where the LVF dies out (Figure 1 & Figure 4a). Consequently, similar thickness increases of lower Carboniferous and upper Carboniferous sedimentary strata elsewhere within graben and half-graben structures on the Finnmark Platform east suggest that syn-tectonic sediment deposition

occurred in Carboniferous times along the LVF and analog ENE-WSW to NNE-SSW trending faults. Furthermore, in the footwall of the northern fault segments of the MFC, we recorded anomalously thick upper Carboniferous succession (Figure 8b) with a thickness comparable to what is observed within the southwesternmost Nordkapp basin (Figure 8b). This succession shows a half-ellipsoid shape in map-view with a NE-SW trending major axis parallel to the MFC (Figure

8b). We therefore argue that this thickness change on the Finnmark Platform east is the result of syn-tectonic sediment deposition in the hanging-wall of a NE-SW trending, SE-dipping fault antithetic to the MFC (Figure 4a & b). We suggest that the half-ellipsoid shape of the thickened upper Carboniferous sedimentary deposits on the Finnmark Platform east reflects large offset near the center of a the SE-dipping fault and decreasing vertical throw towards the fault-tips, a feature

characterizing syn-sedimentary, rift-related normal faults (Figure 8b).

By contrast, depositional sediment wedges may as well occur on the Finnmark Platform east, and they differ from fault-controlled thickness changes. One example is the ca. 600 m-thick lower Carboniferous succession evidenced by shallow drilling between the Nordkinn Peninsula and Magerøya (cf. "star" symbol in Figure 1; Bugge et al., 1995), which we re-interpreted as a

prograding Carboniferous sedimentary system (Figure 5d). The apparent thickening of the lower Carboniferous succession near the coasts of NW Finnmark is more likely to be related to sedimentary processes during the formation of large clinoforms in a prograding sedimentary system (Figure 5d) than to syn-tectonic deposition in the hanging-wall of a NE-SW trending, NW-dipping fault.




*Fault-controlled thickness changes in Permian strata*

In the southwesternmost Nordkapp basin and on the Finnmark Platform east and west, the Permian sedimentary succession is thin and shows a relatively constant thickness compared to the underlying Devonian-lower Carboniferous and upper Carboniferous successions (Figure 4a-d and

Figure 8a-c). However, the Base Asselian and Base Triassic reflections marking the lower and upper boundary of the Permian succession show some offsets across the Rolvsøya fault, thus accounting for minor thickness variations in the Permian succession across this fault (Figure 4c & d and Figure 8c). This suggests that the main tectonic activity along the Rolvsøya fault was essentially restricted to the Mid/Late Devonian-late Carboniferous. Moreover, on the Finnmark

Platform west and east, most brittle faults die out within the upper Carboniferous succession and only a few faults crosscut the Permian succession with limited amount of offset (Figure 4a, b, e, & f). Across the TFFC and the MFC, the Base Asselian and Base Triassic reflections display small offsets, which we interpret as mild Mesozoic reactivation of these faults (Figure 4c, d & g).

*Fault-controlled thickness changes in Mesozoic-Cenozoic strata*

Most faults observed within the late Paleozoic succession on the Finnmark Platform east and west and in the southwesternmost Nordkapp basin die out in the upper part of the succession before reaching the Base Triassic reflection (Figure 4). A few exceptions exist where the MFC and the WNW-ESE trending segment of the TFFC show small offsets of Mesozoic sedimentary strata

(Figure 4c-g). The weak influence of these faults compared to offsets observed within late Paleozoic successions (Figure 4c-g) suggests that, at least some major faults were mildly reactivated in Mesozoic times but, in general, most brittle faults on the Finnmark Platform east and west and in the southwesternmost Nordkapp basin remained inactive after Carboniferous times.

**4.2.    Offshore aeromagnetic data**

To better verify our 2D interpretation of faults and basin architectures on the Finnmark Platform and in the southwesternmost Nordkapp basin, we compare and tie our results using high-resolution, offshore aeromagnetic data from Gernigon et al. (2014; Figure 3). Aeromagnetic



anomalies, when combined with seismic interpretation, may provide useful results allowing to identify brittle faults and offset patterns (cf. Indrevær et al., 2013).

On the Finnmark Platform east, offshore aeromagnetic data (Figure 3; Gernigon et al., 2014) show multiple narrow, NNE-SSW trending, high-positive aeromagnetic anomalies that bend into NW-SE/NNW-SSE orientations near the center of the Nordkapp Basin, which Gernigon et al.

(2014) interpreted as arc-shaped prolongations of Caledonian nappes. A more detailed analysis of these aeromagnetic data reveals a set of triangular to rhomboidal, high-negative aeromagnetic anomalies, the largest of which was observed northeast of the island of Magerøya (dashed white lines in Figure 3). This high-negative anomaly is bounded to the northeast and to the northwest by narrow, linear, NNE-SSW to NE-SW trending, high-positive aeromagnetic anomalies (dashed

white lines in Figure 3). On seismic data, the locations of these linear, high-positive aeromagnetic anomalies coincide with SE-dipping normal fault for the northwestern anomaly, and the NW-dipping, zigzag-shaped LVF for the southeastern anomaly (cf. black arrows in Figure 4a & b). These two faults bound a triangular-shaped basin filled up with thickened Carboniferous sedimentary deposits (cf. Figure 4a & b), which shape and extent mimic those of the triangular,

high-negative anomaly observed on aeromagnetic data northeast of Magerøya (Figure 3). Such triangular-shaped, high-negative aeromagnetic anomaly may thus be indicators of offshore Carboniferous sedimentary basins.

Similarly, on the Finnmark Platform west, a large NE-SW trending, linear, high-positive aeromagnetic anomaly is observed in the footwall of the TFFC (dotted white lines in Figure 3),

where it extends northeastwards into the footwall of the Rolvsøya fault (Figure 3). This NE-SW trending, high-positive aeromagnetic anomaly coincides with a NE-SW trending basement ridge in the footwall of the TFFC on the Finnmark Platform west and with the location of an ENE-WSW trending basement ridge in the footwall of the Rolvsøya fault (Figure 1 and Figure 4c-f). We interpret this NE-SW trending, high-positive aeromagnetic anomaly to highlight a significant

compositional difference between highly-magnetic basement rocks in NE-SW and ENE-WSW trending basement ridges and poorly magnetic, adjacent basement rocks on the Finnmark Platform west and in the southwesternmost Nordkapp basin (Figure 4c-f).

## 5.  Discussion




Our regional and detailed seismic studies of basin-boundary faults such as the TFFC, MFC, Rolvsøya fault and TKFZ on the Finnmark Platform and adjacent southwesternmost Nordkapp basin show multiple links and interactions. We focus the discussion on the interaction of these faults and associated minor faults on Late-Devonian-Carboniferous (half-) graben basins. We

specifically discuss how deep-seated ductile Caledonian shear zones, i.e. the Sørøya-Ingøya shear zone and basement ridges may have been exhumed and thus enabled to control post-Caledonian brittle faulting and formation of Late-Devonian-Carboniferous basins as collapse basins. In combinaison, the structural architecture, timing of faulting and fault-controlled thickness variations on the Finnmark Platform and in the southwesternmost Nordkapp basin provide the framework to

discuss the evolution of the SW Barents Sea margin from the Mid/Late Devonian to the Permian.

## 5.1.   Interaction of the main segment of the Måsøy Fault Complex with the Sørøya-Ingøya shear zone

The linear, NE-SW trending geometry of the main segment of the MFC in map-view (Figure 1 & Figure 2) strongly differs from the dominant ENE-WSW to NNE-SSW trending, zig-zag pattern typically observed for post-Caledonian faults in Mid-Norway (Blystad et al., 1995), Lofoten-Vesterålen (Bergh et al., 2007; Eig, 2008; Hansen et al., 2012), Western Troms (Indrevær et al., 2013) and NW Finnmark (Koehl et al., submitted). Notably, the anomalously linear fault

segment of the MFC trends fully parallel to and soles into high-amplitude, NW-dipping seismic reflections of the SISZ on the Finnmark Platform and in the southwesternmost Nordkapp basin (Figure 4c-f). This obvious merging of main segment of the MFC into the basement-seated SISZ (Figure 4c-f) suggests it formed as a brittle splay fault along an inverted portion of the SISZ, likely during the collapse of the Caledonides in the Mid/Late Devonian (Gudlaugsson et al., 1998). We

suggest a similar interpretation for the Rolvsøya fault, which also flattens and merges into a bowed portion of the SISZ (Figure 4c & d), and for the northwest-boundary fault of the Devonian graben on the Finnmark Platform west soling into a minor, SE-dipping shear zone (Figure 4e & Figure 5b & c). These faults are thought to have remained active through the late Carboniferous as suggested by potential syn-tectonic sediment thickening within the upper Carboniferous succession (Figure

8b) but most likely ceased before the Permian as supported by the relatively constant thickness of Permian sedimentary strata throughout the study area (Figure 8c).



By analogy, in the central North Sea, Phillips et al. (2016) successfully tied the southernmost onshore occurrence of the Karmøy Shear Zone, a major Caledonian shear zone, to a thick seismic unit made up with sub-parallel, high-amplitude reflections similar to those ascribed

to the SISZ in the footwall of the main segment of the MFC (Figure 4d-f). Phillips et al. (2016) argue that the Åsta Fault, a large N-S trending, W-dipping, post-Caledonian fault in the North Sea, formed during a phase of extensional reactivation of the Karmøy Shear Zone. Similarly, in western Norway, Wilks & Cuthbert (1994) proposed that the Hornelen Basin formed along a brittle fault that splayed upwards from the Nordfjord-Sogn Detachment Zone during Middle Devonian late-

orogenic extension.

### 5.2.      Formation of the WNW-ESE trending fault segment of the Troms-Finnmark Fault Complex as a hard-linked accommodation cross-fault

Our data (Figure 8a & b) show abrupt fault-controlled thickening of the Devonian-lower Carboniferous and upper Carboniferous sedimentary successions just northeast of the WNW-ESE trending segment of the TFFC into the southwesternmost Nordkapp basin. On the Finnmark Platform west, potential Devonian sedimentary rocks are truncated upwards by the mid-Carboniferous reflection, which we interpret as a major angular unconformity (Figure 4e & f). This

unconformity may have been caused by a major eustatic sea-level fall in mid-Carboniferous (Serpukhovian) times that exposed large areas to coastal erosion (Saunders & Ramsbottom, 1986). We propose, for example, that the absence of high-amplitude, coal-bearing sedimentary deposits of the Billefjorden Group (lower Carboniferous) on the Finnmark Platform west is related to this major episode of eustatic sea-level fall in the Serpukhovian, which may have contributed to expose

lower Carboniferous sedimentary rocks in this area to coastal erosion. Hence, part of the thickening of the Devonian-lower Carboniferous succession across the WNW-ESE trending segment of the TFFC might be related to extensive erosion of the Finnmark Platform west in mid-Carboniferous times. In addition, the clear deepening (plunge) to the northeast of the spoon-shaped trough formed by the three-dimensionally folded and bowed geometry of the SISZ (Figure 6) suggests that the

thickening of Devonian-lower Carboniferous sedimentary strata into the southwesternmost Nordkapp basin (Figure 8a) is also partly controlled by the shape and attitudes of the underlying SISZ. Finally, the thickened sediment depocenter observed in the southwesternmost Nordkapp



basin at the intersection of the TFFC and the MFC (Figure 8a & b) is at least partly related to syn-sedimentary normal faulting along the WNW-ESE trending fault segment of the TFFC and along

the main segment of the MFC. This most likely indicates that the TFFC and the MFC had already merged and acted as a single fault zone during sediment deposition in the southwesternmost Nordkapp basin from the end of the Serpukhovian and potentially from Devonian times. We propose that the WNW-ESE trending fault segment of the TFFC acted as an accommodation cross-fault, as defined in Sengör (1987), that transferred displacement between the NNE-SSW trending

segment of the TFFC and the main segment of the MFC, defining a step synthetic with the deepening direction of the spoon-shaped trough formed by the geometry of the SISZ (Figure 6). This interpretation is based on the dominant dip-slip kinematic of the WNW-ESE trending segment of the TFFC and on its sub-parallel strike to the dominant WNW-ESE trending extension direction inferred along the SW Barents Sea margin during late Paleozoic times (Bergh et al., 2007; Eig &

Bergh, 2011; Hansen & Bergh, 2012). Further, we infer that the strike and location of the WNW-ESE trending segment of the TFFC was controlled by the geometry of the underlying SISZ (see below), which dips gently to the northeast on the Finnmark Platform west and in the southwesternmost Nordkapp basin and may therefore have favorized the formation of a NE-dipping fault at this location (Figure 4g & Figure 6).

Alternatively, Lea (2016) proposed that the WNW-ESE trending fault segment of the TFFC corresponds to a breached relay-ramp fault between the NNE-SSW trending fault segment of the TFFC and the MFC. However, this model implies that this portion of the TFFC would have accommodated significantly fewer displacement than the two faults it links (i.e. the NNE-SSW trending segment of the TFFC and the MFC), which is clearly not the case. The offset of the mid-

Carboniferous reflection and the thickness increase both of the Middle/Upper Devonian-lower Carboniferous and upper Carboniferous sedimentary successions across the WNW-ESE trending segment of the TFFC are comparable to the offset and thickness increase observed across the main fault segment of the MFC (Figure 4c, d & g).

**5.3.   Devonian collapse basins on the Finnmark Platform west (Gjesvær Low) and in the southwesternmost Nordkapp basin**



Our seismic interpretation has shown that Devonian sedimentary rocks in the SW Barents Sea may exist on the Finnmark Platform west and in the southwesternmost Nordkapp basin (Figure 4c-g). The most probable occurrence is on the Finnmark Platform west, in the hanging-wall of the main segment of the MFC (Figure 1, Figure 4e). The presumed Devonian seismic unit corresponds to a suite of low-amplitude reflections crosscut by a few, moderate-amplitude reflection that dip gently to the northeast (Figure 4e). The main argument for a Devonian sequence is that these reflections are remarkably different from the typical seismic patterns observed for lower Carboniferous sedimentary deposits and basement rocks. Lower Carboniferous sedimentary deposits of the Billefjorden Group are characterized by high-amplitude reflections produced by coal-bearing sedimentary rocks (Figure 4a & b and Figure 5a), while basement rocks are mostly associated with thick packages of chaotic seismic reflections (Figure 5a-c) and thick layers of moderate- to high-amplitude, sub-parallel seismic reflections that we interpreted as basement-seated shear zones (e.g. the SISZ; Figure 5f). Another argument in favor of Devonian sedimentary deposits is the presence of a NE-SW trending gravimetric low on the Finnmark Platform west: the Gjesvær Low (Johansen et al., 1994). Devonian sedimentary rocks in Svalbard show an average density of ca. 2.4 g.cm$^{-3}$ associated to depths of 0-8 km (i.e. average depth of 4 km; Manby & Lyberis, 1992), which is less dense than metamorphosed Caledonian rocks (2.6-3.0 g.cm$^{-3}$) and Carboniferous sedimentary rocks on the Finnmark Platform west (< 2.5 g.cm$^{-3}$; Johansen et al., 1994). However, taking into account the effect of burial up to a depth of 5-6 km on the Finnmark Platform (Johansen et al., 1994) and the resulting density increase for Devonian sedimentary deposits with an approximate rate of ca. 0.15 g.cm$^{-3}$.km$^{-1}$ (cf. "all rocks density-depth gradient" in table 3 in Maxant 1980), Devonian sedimentary rocks on the Finnmark Platform may reach densities 2.55-2.7 g.cm$^{-3}$. Thus, the occurrence of the Gjesvær low can be explained by the presence of intermediate-density Devonian sedimentary rocks below the mid-Carboniferous reflection (Figure 4e & Figure 5b & c). This is as well in accordance with density variations and related estimates of Johansen et al. (1994) in the Gjesvær low.

In addition, the discrete, moderate-amplitude, NW-dipping reflections observed within the presumed Devonian sedimentary strata on the Finnmark Platform west may represent syn-tectonic sedimentary growth strata (Figure 4e & Figure 5b & c). These strata are located above and thickened against arcuate, high-amplitude reflections that we interpreted as a major erosional unconformity that truncates SE-dipping Caledonian basement shear zones subparallel to the SISZ



(dotted yellow lines in Figure 5b & c). We consider the Devonian sedimentary rocks on the

Finnmark Platform west to have been deposited in wedge-shaped late/post-Caledonian extensional

basins due to reactivation of a set of partly eroded, exhumed, SE-dipping Caledonian shear zones

(dotted yellow lines in Figure 4e & Figure 5b & c). In mid-Norway, Braathen et al. (2000) reported

a similar setting of Middle Devonian sedimentary basins located above Caledonian shear zones

and folded nappe stack, and proposed that Middle Devonian basins in western and mid-Norway

formed during extensional reactivation of the Caledonian shear zones. Such a model is further

supported by the similar geometry of the Devonian sedimentary growth strata on the Finnmark

Platform west to the highly tilted geometry of sedimentary strata in Middle Devonian basins in

western Norway (cf. Séranne & Seguret, 1987; Séranne et al., 1989; Wilks & Cuthbert, 1994;

Osmundsen & Andersen, 2001). Moreover, the Devonian sedimentary basins on the Finnmark

Platform west (Figure 4e and Figure 5b & c) and in the southwesternmost Nordkapp basin (Figure

4c & d) define NE-SW trending graben structures with comparable, < 50 km-wide sizes to those

of the Middle Devonian Hornelen, Kvamsheten and Solund basins in western Norway (Séranne &

Seguret, 1987; Osmundsen & Andersen, 2001).

In the southwesternmost Nordkapp basin, the presence of Middle/Upper Devonian

sedimentary rocks is more speculative and is mostly based on the maximum thickness of lower

Carboniferous sedimentary deposits registered in the SW Barents Sea, which is ca. 600 m on the

Finnmark Platform east (Bugge et al., 1995; Figure 5d). Assuming a seismic velocity < 6 km.s$^{-1}$

for lower Carboniferous coal-bearing sedimentary deposits, a thickness of 600 m would account

for only part (max. 0.2 s) of the 2-2.5 second-thick (TWT) seismic unit observed below the mid-

Carboniferous reflection in the southwesternmost Nordkapp basin (Figure 4c & d). If basement

rocks were present at the base of the southwesternmost Nordkapp basin, they would most likely

produce a seismic reflection pattern similar to that of the sub-parallel, high-amplitude reflections

of the underlying SISZ (Figure 4c & d) or potentially form an unconformity to the overlying late

Paleozoic sedimentary rocks. We therefore believe that the southwesternmost Nordkapp basin is

bounded below by the SISZ, which gives the basin a peculiar "U" shape in cross-section (Figure

4c & d) and composed of thick Middle/Upper Devonian sedimentary deposits overlain by lower

Carboniferous sedimentary strata below the mid-Carboniferous reflection. From these arguments,

i.e. brittle extensional reactivation, bowed geometry and controlling effect of the basement-seated

SISZ (Figure 6), we suggest deposition of Devonian sedimentary rocks within a late/post-



Caledonian, spoon-shaped, collapse basin formed along bowed, inverted portions of the
Caledonian SISZ (Figure 4c-f), thus representing analogs to Middle Devonian collapse basins in
western and mid-Norway (Séranne et al., 1989; Wilks & Cuthbert, 1994).

## 5.4.     Formation of NE-SW to ENE-WSW trending basement ridges as exhumed
metamorphic core complexes

We have argued for an upward-bowed seismic geometry of the SISZ (Figure 4c-fFigure 6)
into which major fault complexes such as the TFFC, the MFC and the Rolvsøya fault merge and
sole into (Figure 4c-f). In map-view (Figure 6) and cross-section (Figure 4c-f), the bowed geometry
of the SISZ defines two, ENE-WSW to NE-SW trending ridges of basement rocks on the
northwestern flanks of presumed Devonian basins. These basement ridges correlate well by
displaying high-positive gravimetric (fig. 5 in Olesen et al., 2010 and fig. 5 in Gernigon et al.,
2014) and high-positive aeromagnetic anomalies (Figure 3; Gernigon et al. 2014) that suggest these
ridges are made of basement lithologies significantly different from adjacent basement rocks on
the Finnmark Platform west and in the southwesternmost Nordkapp basin. These basement ridges
seem to align with high-positive gravimetric anomalies that coincide with the NE-SW trending
Norsel High (Gabrielsen et al., 1990) along the northwestern flank of the Nordkapp Basin in the
northeast. Farther southwest, these basement ridges coincide with the NE-SW trending West Troms
Basement Complex in Western Troms (Bergh et al., 2010) and the NE-SW trending Lofoten Ridge
in Lofoten-Vesterålen (Bergh et al., 2007), among which at least the Lofoten Ridge was exhumed
as a metamorphic core complex (Klein & Steltenpohl, 1999; Klein et al., 1999; Steltenpohl et al.,
2004; 2011) along inverted Caledonian shear zones such as the Eidsfjord and Fiskefjord shear
zones (Steltenpohl et al., 2011). By comparison, the SISZ seems to coincide with high-positive
aeromagnetic anomalies on the Finnmark Platform west that follow the trace of the MFC (Indrevær
et al., 2013) and continue past the southwestern fault-tip of the MFC (Gernigon & Brönner, 2012).
The aeromagnetic anomalies visible on the dataset Gernigon & Brönner (2012) appear to line up
with aeromagnetic anomalies onshore the island of Vannøya, in the northeasternmost part of the
West Troms Basement Complex, and these onshore anomalies correlate with NE-SW trending, SE-
dipping basement shear zones that were reactivated as extensional brittle faults (Paulsen et al. pers.
comm.; Figure 1).



These data indicate that SE-dipping portions of the SISZ propagated southwest of the MFC fault-tip on the Finnmark Platform west and possibly merged onshore Vannøya in Western Troms with a suite of NE-SW trending, SE-dipping shear zones. As a consequence, the basement ridges observed on the FinnmarkPlatform west and along the northern flank of the southwesternmost

Nordkapp basin may have formed as part of a large metamorphic core complex, which included the Lofoten Ridge, the West Troms Basement Complex and possibly also the Norsel High (Figure 1), exhumed along inverted Caledonian shear zones such as the SISZ on the SW Barents shelf and the analogous Eidsfjord and Fiskefjord shear zones in Lofoten-Vesterålen (Steltenpohl et al., 2011).

The timing, nature of uplift and processes of core complex exhumation can be inferred from

thickness variations of the SISZ in cross-section, for example, thickest in the footwall of the MFC and TFFC and thinnest below these two fault complexes (Figure 4f). We link these thickness variations to excisement and incisement processes (cf. Lister & Davis, 1989) along the SISZ during core complex exhumation, after the embrittlement of the SISZ (Figure 9). A model of Devonian late/post-orogenic extension is proposed, when the SISZ was inverted as a low-angle extensional

detachment that contributed to crustal thinning through top-to-the-NW tectonic transport (Figure 9a). Relatively rapid thinning above the upper part of the SISZ may have triggered early exhumation of basement rocks on the Finnmark Platform west and along the northern flank of the southwesternmost Nordkapp basin (Figure 9a), causing the upper part of the SISZ to bow upwards (Figure 9b). Continued crustal extension and exhumation of basement rocks below the upper part

of the SISZ led the bowed portion of the SISZ to become unsuitably oriented to accommodate top-to-the-NW extensional displacement and therefore become inactive (dashed black line in Figure 9c). Further extension likely led the SISZ to splay upwards into its hanging-wall, becoming suitable again to accommodate top-to-the-NW extension displacement (Figure 9c). This upward splay-faulting process is referred to as excisement by Lister & Davis (1989) and we tentatively apply this

process to explain the observed thickening of the SISZ in the footwall of the MFC (Figure 4f). Further crustal thinning along the SISZ may have triggered the exhumation of basement rocks along progressively deeper parts of the SISZ (Figure 9b & c), causing bend-up of the SISZ at even deeper crustal levels (Figure 9d). Extreme bowing of lower portions of the SISZ led to opposite top-to-the-SE transport direction on the Finnmark Platform west and in the southwesternmost

Nordkapp basin (Figure 9d), which contributed to exhume NE-SW to ENE-WSW trending ridges of basement rocks in the footwall of the NNE-SSW trending segment of the TFFC (Figure 4e & f)



and in the footwall of the Rolvsøya fault (Figure 4c & d). Incisement (downward splaying; cf. Lister & Davis, 1989) may have occurred below the basement ridges during continued top-to-the-NW extension along the SISZ (Figure 9e) and possibly contributed to thicken the SISZ in the

footwall of the TFFC (Figure 4e & f) and of the Rolvsøya fault (Figure 4c & d), resulting in the current geometry of the SISZ (Figure 9f). We believe that extreme bowing of lower portions of the SISZ and associated local top-to-the-SE displacement contributed to form the spoon-shaped trough in which Devonian sedimentary rocks are thought to be deposited on the Finnmark Platform west and in the southwesternmost Nordkapp basin (Figure 6 & Figure 9d & e).

By comparison, in northeast Greenland, Sartini-Rideout et al. (2006) and Hallett et al. (2014) proposed that ultra-high pressure (UHP) basement rocks were exhumed along large, mylonitic, Caledonian shear zones in Late Devonian-early Carboniferous times (ca. 370-340 Ma). The study of Sartini-Rideout et al. (2006) also shows that the last stages of exhumation were accommodated by steep, brittle normal faults that strike parallel to major Caledonian shear zones,

i.e. similar to the main segment of the MFC that strikes parallel to the SISZ along the SW Barents Sea margin (Figure 1). In addition, results from sediment provenance and geochronological studies by McClelland et al. (2016) in Carboniferous basins in northeast Greenland showed that the exhumation of UHP basement rocks as elongated ridges could have formed a regional Serpukhovian unconformity, i.e. contemporaneous with the mid-Carboniferous (Serpukhovian?)

unconformity observed on the Finnmark Platform east (Figure 4a & b and Figure 5a; Bugge et al., 1995) and on the Finnmark Platform west (Figure 4e-g & Figure 5b & c). These observations are compatible with the results of Saunders & Ramsbottom (1986) who argued that major eustatic sea-level fluctuations occurred in Serpukhovian times and that these fluctuations included an episode of sea-level fall and continental erosion around 330 Ma in the early Serpukhovian, and an episode

of sea-level rise at ca. 325 Ma at the end of the Serpukhovian. In late Paleozoic times, the northeast Greenland margin was located close to its conjugate counter-part of the SW Barents Sea margin and these two areas were most likely subjected to similar regional stresses and closely related sea-level fluctuations. Therefore, we suggest that the mid-Carboniferous unconformity reflection observed in the SW Barents Sea (cf. Figure 4 & Figure 5a-c; Bugge et al., 1995), formed as a

response to major eustatic sea-level fall in the early Serpukhovian (Saunders & Ramsbottom, 1986) and due to large-scale exhumation of basement rocks in Late Devonian-early Carboniferous times. Exhumation occurred along inverted Caledonian shear zones (e.g. SISZ) and brittle splay faults



such as the main segment of the MFC, the NNE-SSW trending segment of the TFFC, the Rolvsøya fault and the NNE-SSW trending, SE-dipping fault that bounds potential Devonian sedimentary strata on the Finnmark Platform west to the northwest (Figure 4). The timing of exhumation can be constrained to the end of the Serpuhkovian based on deposition of thick alluvial and shallow marine upper Carboniferous sedimentary deposits of the Gipsdalen Group on top of the mid-Carboniferous unconformity in the SW Barents Sea (Figure 4), associated with an eustatic sea-level rise in the end of the Serpukhovian (Saunders & Ramsbottom, 1986).

In Lofoten-Vesterålen, Steltenpohl et al. (2004) inferred a Late Devonian age for the exhumation of metamorphic core complexes and this age was refined by recent $^{40}$Ar/$^{39}$Ar isotopic results (Steltenpohl et al., 2011), which constrained extensional reactivation of the Eidsfjord shear zone to the Early Devonian. In the SW Barents Sea, much work is needed to better constrain the timing of late/post-Caledonian extension and collapse basin formation. Nonetheless, we believe that the Early Devonian age obtained in Lofoten-Vesterålen (Steltenpohl et al., 2011) represents a reasonable estimate for the onset of crustal thinning in the SW Barents Sea (Figure 9a-c). Additionally, Late Devonian-early Carboniferous timing of exhumation for basement rocks in northeast Greenland and formation of a regional mid-Carboniferous (Serpukhovian) unconformity (Sartini-Rideout et al., 2006; Hallett et al., 2014; McClelland et al., 2016) corresponds to a realistic approximation for the final stages of late/post Caledonian extension, ending with the formation of Devonian-Carboniferous collapse basins along exhumed, NE-SW to ENE-WSW trending basement ridges (Figure 9d-f).

### 5.5. Interaction of the Trollfjord-Komagelv Fault Zone with the Troms-Finnmark and Måsøy fault complexes

The prolongation of the TKFZ from onshore areas in eastern Finnmark to offshore areas of the SW Barents Sea has been a matter of debate. Most studies tend to connect the onshore TKFZ with the offshore WNW-ESE trending fault segment of the TFFC (Gabrielsen, 1984; Gabrielsen & Færseth, 1989; Roberts et al., 2011; Bergø, 2016; Lea, 2016). Our data, however, suggest that the TKFZ dies out near the coasts of NW Finnmark (present contribution and Koehl et al., submitted), and in this section we review and discuss new evidence obtained from the interpretation of offshore seismic and aeromagnetic data.



First, the TKFZ was described onshore eastern Finnmark as a major sub-vertical fault that

accommodated dominantly strike-slip movement (Roberts, 1972; Rice, 2013). Farther west, the TKFZ crops out onshore Magerøya, where it is made of numerous, high-frequency, subparallel, subvertical, WNW-ESE trending faults and fractures that accommodated at least small-scale post-Caledonian, strike-slip to oblique-slip displacement (Koehl et al., submitted). By contrast, seismic interpretation of the WNW-ESE trending fault segment of the TFFC shows that this fault exhibits

a typical, high-angle (ca. 70°) normal fault geometry and accommodated significant amount of post-Caledonian normal dip-slip displacement (Figure 4g). Thus, the geometries of the TKFZ and of the WNW-ESE trending segment of the TFFC contrast significantly with each other. Second, the imaginary prolongation of the TKFZ from the island of Magerøya to the WNW, onto the Finnmark Platform, would crosscut the Finnmark Platform west nearly 23 km southwest of the

observed trace of the WNW-ESE trending segment of the TFFC (Figure 4g). This represents an significant mismatch that is too important to represent minor dextral strike-slip offset of the TKFZ across the main fault segment of the MFC, which dominantly accommodated normal dip-slip motions (Figure 4c-f). Third, the interpretation of 3D seismic data at the intersection of the MFC and TFFC reveals that the footwall of the MFC is largely intact or seismically unaffected (Figure

7). There is rather not any evidence of intense fracturing as typically observed along the TKFZ onshore Magerøya (Koehl et al., submitted). We therefore believe that the TKFZ and the WNW-ESE trending fault segment of the TFFC represent two distinct faults. These data more likely suggest that the TKFZ dies out instead of propagating onto the Finnmark Platform, and that the WNW-ESE trending fault segment of the TFFC does not continue through the MFC. This

interpretation is supported by the absence of other WNW-ESE trending offshore faults, such as the Austhavet fault previously interpreted near the coasts of Finnmark (Townsend, 1987b; Lippard & Roberts, 1987; Roberts et al., 2011). We re-interpret the Austhavet fault as seismic artifacts related to the Djuprenna trough, a large glacial trough that trends parallel to the northeastern coast of Finnmark (Ottesen et al., 2008; Rise et al., 2015). Such a conclusion is supported by shallow

drillings on the Finnmark Platform east, showing no sign of fault-related offset on this part of the Finnmark Platform east (cf. fig. 4 Bugge et al., 1995). Similarly, our mapping and regional analysis of brittle faults on the Finnmark Platform show very few occurrences of WNW-ESE trending faults (Figure 1 & Figure 2).

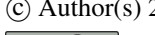



Our onshore studies (Koehl et al., submitted) show an increased number of large-scale
WNW-ESE trending fault segments and splays along the TKFZ, varying from a single-segment
fault onshore the Varanger Peninsula in eastern Finnmark (Figure 1) to multiple segments onshore
and nearshore Magerøya. This suggests that the island of Magerøya is located near the fault-tip
process zone (Shipton & Cowie, 2003; Braathen et al., 2013) of the TKFZ, and that the TKFZ
therefore dies out to the west before reaching the Finnmark Platform and the southwesternmost
Nordkapp basin. Nearby Magerøya and the Nordkinn Peninsula, high-resultion aeromagnetic data
reveal the presence of highly magnetic dolerite dykes along WNW-ESE trending fault segments of
the TKFZ (Roberts et al., 1991; Nasuti et al., 2015; Koehl et al., submitted). These narrow, high-
positive aeromagnetic anomalies also die out westwards (Gernigon & Brönner, 2012; Gernigon et
al., 2014), therefore supporting that the dolerite dykes and, thus, the TKFZ may die out before
reaching the Finnmark Platform. We consider an alternative model to the fault-tip process zone of
Koehl et al. (submitted). We argue that, during the Caledonian Orogeny, the Precambrian, WNW-
ESE trending TKFZ was oriented subparallel to the dominant top-to-the-SE transport direction of
Caledonian thrusts in northern Norway (Townsend, 1987a) and thus not suitably oriented to be
reactivated as a major thrust or strike-slip fault. Instead, our data indicate that if the TKFZ extended
farther west onto the Finnmark Platform west prior to the onset of Caledonian deformation and was
truncated and decapitated by the SISZ and associated NE-SW trending, Caledonian thrusts and
shear zones during top-to-the-SE thrusting, the western continuation of the TKFZ on the Finnmark
Platform may have been transported/thrusted southeastwards along the SISZ and is now most likely
eroded. However, portions of the western prolongation of the TKFZ may have been preserved
offshore in basement highs such as the Loppa and Veslemøy highs (Figure 1), assuming that the
TKFZ extended west of Magerøya into the SW Barents Sea prior to the Caledonian Orogeny. This
is supported by the recent observation of subvertical, WNW-ESE trending brittle faults on the
Veslemøy High (Kairanov et al., 2016).

**5.6.     Late Paleozoic evolution of the SW Barents Sea margin**

Based on the seismic data and discussions from previous chapters we now address the
tectonic evolution of the Finnmark Platform and the southwesternmost Nordkapp basin in the late
Paleozoic (Figure 10). The main structural element discussed in our model is the SISZ and we link



its influence on (i) the development of the southwesternmost Nordkapp basin, (ii) the geometry of
the TFFC and MFC, (iii) the deposition of Mid/Late Devonian sedimentary rocks in the
southwesternmost Nordkapp basin and on the Finnmark Platform west, (iv) transfer faults such as
the TKFZ and (v) the deposition of syn-tectonic sedimentary wedges along steep normal faults that
bound triangular-shaped basins in the Carboniferous.

We have demonstrated the presence of major, basement-seated, Caledonian shear zones on
the Finnmark Platform west and below the southwesternmost Nordkapp basin. The largest of these
shear zones is the SISZ, which strikes NE-SW and dips to the northwest (Figure 4c-g & Figure 5f).
The trend and dominant northwestern dip of this shear zone suggest that it formed as a large thrust
that accommodated top-to-the-SE tectonic transport during the Caledonian Orogeny. The SISZ has

a bow-shaped, three-dimentionally folded geometry that coincides with basement ridges in the
footwall of the TFFC and of the Rolvsøya fault (Figure 4c-f, Figure 6 and Figure 10). We propose
that the SISZ and potential other Caledonian shear zones along the SW Barents Sea margin were
inverted as low-angle extensional shear zones during late/post-Caledonian orogenic extension and
subsequent collapse. This is based on analog examples in northeast Greenland (Sartini-Rideout et

al., 2006; Hallett et al., 2014; McClelland et al., 2016), western Norway (Séranne & Seguret, 1987;
Séranne et al., 1989; Wilks & Cuthbert, 1994; Osmundsen & Andersen, 2001), mid-Norway
(Braathen et al., 2000) and Lofoten-Vesterålen (Klein & Steltenpohl, 1999; Klein et al., 1999;
Steltenpohl et al., 2004; 2011; Osmundsen et al., 2005). Extensional reactivaton of such ductile
shear zones along the Barents Sea margin might have initiated in the Early Devonian as in Lofoten-

Vesterålen (Steltenpohl et al., 2011), by crustal thinning and orogenic collapse through dominant
top-to-the-NW displacement along the SISZ and later exhumation of the SISZ and underlying
basement as a metamorphic core complex. Reactivation of the exhumed basement ridges occurred
by normal faulting along new, steep, brittle faults such as the main segment of the MFC and the
NNE-SSW trending fault segment of the TFFC (cf. Figure 4f), likely due to incisement and

excisement processes (Figure 9; Lister & Davis, 1989). These processes contributed to a
progressive exhumation of basement rocks as ENE-WSW and NE-SW trending ridges along
bowed portions of the SISZ (cf. Figure 4c-f and Figure 10a & b). We believe that these ridges were
part of a larger-scale NE-SW trending metamorphic core complex that included the Norsel High
and the two basement ridges located in the footwall of the TFFC and the Rolvsøya fault. Farther

south, this core complex may be linked to the West Troms Basement Complex (Bergh et al., 2010)



and the Lofoten Ridge (Blystad et al., 1995; Figure 10). Such a regional link is favored by the alignement of NE-SW trending, high-positive gravimetric anomalies that characterize these ridges (Olesen et al., 2010; Gernigon et al., 2014). The timing of final core complex exhumation can be constrained to Mid/Late Devonian-early Carboniferous and possibly linked to the regional

Serpukhovian unconformity on the Finnmark Platform (cf. Figure 4a, b, e, f & g and Figure 5a-c; Bugge et al., 1995), in accordance with Sartini-Rideout et al. (2006) and Hallett et al. (2014) in northeast Greenland.

The exhumation of basement ridges as metamorphic core complexes along the inverted SISZ and subsequent normal faulting along the MFC and TFFC created a deep, spoon-shaped

topographic depression on the Finnmark Platform west and in the southwesternmost Nordkapp basin (Figure 4c, d, e & g, Figure 5b & c and Figure 10a). These depressions were filled with thick Devonian clastic deposits analog to those observed in Middle Devonian collapse basins in western Norway (Séranne et al., 1989; Osmundsen & Andersen, 2001), and with lower Carboniferous coal-bearing and clastic sedimentary rocks of the Billefjorden Group deposited unconformably above

Devonian strata (cf. Figure 10b). These collapse basins are also likely responsible for the gravimetric low observed on the Finnmark Platform west: the Gjesvær Low (Figure 3).

On the Finnmark Platform west, the final stages of core complex exhumation and a major phase of eustatic sea-level fall in the Serpukhovian (Saunders & Ramsbottom, 1986) led to extensive erosion of Devonian and lower Carboniferous sedimentary rocks, therefore explaining

the absence of lower Carboniferous sedimentary deposits and the erosional truncation of Devonian sedimentary strata along this part of the margin (Figure 4e-g & Figure 5b & c). On the Finnmark Platform east, lower Carboniferous sedimentary rocks are preserved as minor syn-tectonic sedimentary wedges within small triangular grabens and half-grabens that correlate with aeromagnetic lows (dashed white line in Figure 3). These grabens are bounded by zigzag-shaped,

Late Devonian-Carboniferous normal faults such as the LVF (Figure 4a & b and Figure 5a), which coincide with narrow, high-positive aeromagnetic anomalies (cf. Figure 3 and black vertical arrows in Figure 4a & b). In addition, triangular basins like the graben bounded by the LVF and the southwesternmost Nordkapp basin were partly offset and segmented by WNW-ESE trending transfer faults that accommodated small amount of strike-slip displacement. Examples include the

TKFZ onshore NW Finnmark, which offsets the LVF in a right-lateral fashion (Koehl et al., submitted), and accommodation cross-faults (Sengör, 1987) that accommodated large amount of



orogen-parallel extension through normal dip-slip movement, for example the WNW-ESE trending fault segment of the TFFC (Figure 4g and Figure 8a).

In the late Serpukhovian, a regional episode of eustatic sea-level rise (Saunders & Ramsbottom, 1986) flooded the Finnmark Platform east and west and allowed the deposition of sedimentary rocks of the upper Carboniferous Gipsdalen Group. These rocks occur as syn-tectonic sedimentary wedges that thicken in the hanging-wall of basin-bounding normal faults such as the LVF on the Finnmark Platform east and the main segment of the MFC on the Finnmark Platform west (Figure 4a, b, e & f and Figure 10c). Similarly, in the southwesternmost Nordkapp basin, which may have remained flooded through the entire phase of eustatic sea-level fall and core complex exhumation, thick, partly evaporitic, upper Carboniferous sedimentary rocks were deposited in the basin and these are thickest at the intersection between the TFFC and the MFC (Figure 4c, d & g and Figure 8b). Thus, the thickening of upper Carboniferous strata probably reflects significant syn-sedimentary normal faulting along these two faults, which may have acted as a single fault during in the final stage of extension in the late Carboniferous (Figure 10c).

Most faults on the Finnmark Platform east and west and in the southwesternmost Nordkapp basin appear to die out below the Base Asselian reflection and those that propagate through the Base Asselian reflection show limited amount of offset within Permian and Mesozoic-Cenozoic sedimentary strata (Figure 4). Moreover, the Permian sedimentary succession shows a rather constant thickness through the entire study area (Figure 4 & Figure 8c). Thus, we argue that late/post-Caledonian extensional faulting linked to the collapse of the Caledonides essentially took place in the Mid/Late Devonian-Carboniferous and came to a halt towards the end of the late Carboniferous (Figure 10d). This presumed timing is consistent with recent K/Ar radiometric dating of brittle fault gouges in Western Troms (Davids et al., 2013) and in NW Finnmark (Torgersen et al., 2014; Koehl et al., 2016), as well as with radiometric dating of dolerite dykes in NW Finnmark (Lippard & Prestvik, 1997), eastern Finnmark (Guise & Roberts, 2002) and on the Kola Peninsula in Russia (Roberts & Onstott, 1995), which constrain significant extensional faulting activity onshore northern Norway and adjacent areas in Russia to the Late Devonian-early/mid Permian. Minor reactivation of major fault complexes occurred in the Mesozoic-Cenozoic and are most likely associated with the rifting of the NE Atlantic (Faleide et al., 2008).

## 6. Conclusions



1) The atypically linear, NE-SW trending, main fault segment of the Måsøy Fault Complex formed as a brittle splay of the inverted Caledonian Sørøya-Ingøya shear zone through excisement processes during the collapse of the Caledonides in the Mid/Late Devonian-early Carboniferous and was active until the end of the late Carboniferous.

2) The WNW-ESE trending fault segment of the Troms-Finnmark Fault Complex developed as a hard-linked, accommodation cross-fault that accommodated orogen-parallel, late/post-orogen extension in the Mid/Late Devonian-Carboniferous. This fault merged with the main fault segment of the Måsøy Fault Complex and the two faults acted as a single fault at least during the late Carboniferous, but potentially from Devonian-early Carboniferous times, and accommodated the deposition of thick Devonian-lower Carboniferous and partly evaporitic upper Carboniferous deposits in the southwesternmost Nordkapp basin before faulting came to a halt towards the end of the late Carboniferous.

3) Low-gravity anomalies in the Gjesvær Low and southwesternmost Nordkapp basin may result from the presence of a thick, Mid/Late Devonian, spoon-shaped sedimentary basin that developed along an inverted, bowed portion of the Sørøya-Ingøya shear zone during the collapse of the Caledonides and that display a geometry similar to those of Middle Devonian, late/post-orogenic collapse basins in western and mid-Norway.

4) The ENE-WSW and NE-SW trending basement ridges in the footwall of the Troms-Finnmark Fault Complex and on the northern flank of the southwesternmost Nordkapp basin formed through incisement processes and were exhumed along a bowed portion of the Sørøya-Ingøya shear zone during the collapse of the Caledonides in the Mid/Late Devonian-early Carboniferous. These basement ridges are thought to be part of a large-scale, margin-parallel, NE-SW trending, metamorphic core complex that includes a succession of aligned basement highs such as the Lofoten Ridge, the West Troms Basement Complex and the Norsel High. Core complex exhumation is believed to have stopped by the end of the Serpukhovian when a major eustatic sea-level rise flooded the Finnmark Platform, leading to the deposition of widespread upper Carboniferous sediments.

5) The Sørøya-Ingøya shear zone is thought to have truncated and decapitated Precambrian faults such as the Trollfjord-Komagelv Fault Zone through top-to-the-SE thrusting during

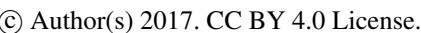


the Caledonian Orogeny and subsequent late/post-orogenic extension. We nevertheless believe that preserved segments of these Precambrian faults might be preserved offshore on basement highs such as the Loppa and Veslemøy highs. However, more work is required in order to map and evaluate the impact of these WNW-ESE trending, subvertical Precambrian faults on the SW Barents Sea margin.

**Data availability**

The seismic data analyzed in this study are part of the DISKOS database and are publicly accessible from any Norwegian academic institution. Aeromagnetic data discussed in the present contribution are from Gernigon et al. (2014).

**Author contribution**

Jean-Baptiste Koehl interpreted the seismic and aeromagnetic data and is the main contributor to the writing process (work-load ca. 45 %). Professor Steffen Bergh provided significant input to the "Introduction" and "Geological Setting" sections as well as detailed critical reviews of the whole manuscript (work-load ca. 30 %). Tormod Henningsen helped initiating the project and provided help with seismic well-ties and regional seismic interpretation (work-load ca. 15 %). Professor Jan-Inge Faleide provided help with the writing process and helped improve the margin evolution model (work-load ca. 10 %).

**Acknowledgements**


The present study is part of the ARCEx project (Research Centre for Arctic Petroleum Exploration), which is funded by the Research Council of Norway (grant number 228107) together with ten academic and eight industry partners. We would like to thank all the persons from these institutions that are involved in this project. We acknowledge the NPD, the NTNU and Schlumberger for sharing seismic data from the DISKOS database, with special thanks to Dicky Harishidayat for taking care of the transfer of the data. We thank Tom Arne Rydningen from the University of Tromsø for reviewing parts of the manuscript. We would also like to thank Hanne-Kristin Paulsen from the University of Tromsø as well as Gwenn Péron-Pinvidic and Per-Terje Osmundsen from the NGU for fruitful discussions.





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



**Figure 1: Regional structural map of the SW Barents Sea margin (based on Bergh et al. 2007, Faleide et al. 2008, Hansen et al. 2012 and Indrevær et al. 2013 and Koehl et al. submitted). The onshore geology is from the NGU and Ramberg et al.**

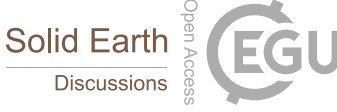



**(2008). The black star marks the location of the speculated half-graben structure described in Bugge et al. (1995), which we reinterpret as a prograding sedimentary system unconformably resting on basement rocks. Abbreviations: AFC = Asterias Fault Complex; Akf = Akkarfjord fault; AW = Alta-Kvænangen tectonic window; BFC = Bjørnøyrenna Fault Complex;**

**BSFC = Bothnian-Senja Fault Complex; BKFC = Bothnian-Kvænangen Fault Complex; FPe = Finnmark Platform east; FPw = Finnmark Platform west; FTZ = Fugløya transfer zone; GL = Gjesvær Low; Ig =Ingøya; KF = Kokelv Fault; L = Langfjorden; Lf = Laksvatn fault; LR = Lofoten Ridge; LVF = Langfjord-Vargsund fault; Ma = Magerøya; Mf = Magerøysundet fault; MFC = Måsøy Fault Complex; NFC = Nysleppen Fault Complex; NH = Norsel High; NP = Nordkinn Peninsula; PP = Porsanger Peninsula; Rf = Rolvsøya fault; RLFC = Ringvassøya-Loppa Fault Complex; RW = Repparfjord-**

**Komagfjord tectonic window; SB = Sørvær Basin; SISZ = Sørøya-Ingøya shear zone; sNB = southwesternmost Nordkapp basin; SSB = Sørøy sub-basin; SSZ = Senja Shear Zone; Sø = Sørøya; TB = Tiddlybanken Basin; TFFC = Troms-Finnmark Fault Complex; TKFZ = Trollfjord-Komagelv Fault Zone; V = Vargsund; Va = Vannøya; VP = Varanger Peninsula; VVFC = Vestfjorden-Vanna fault complex.**

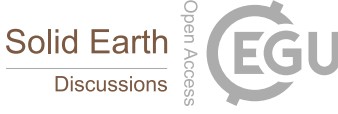



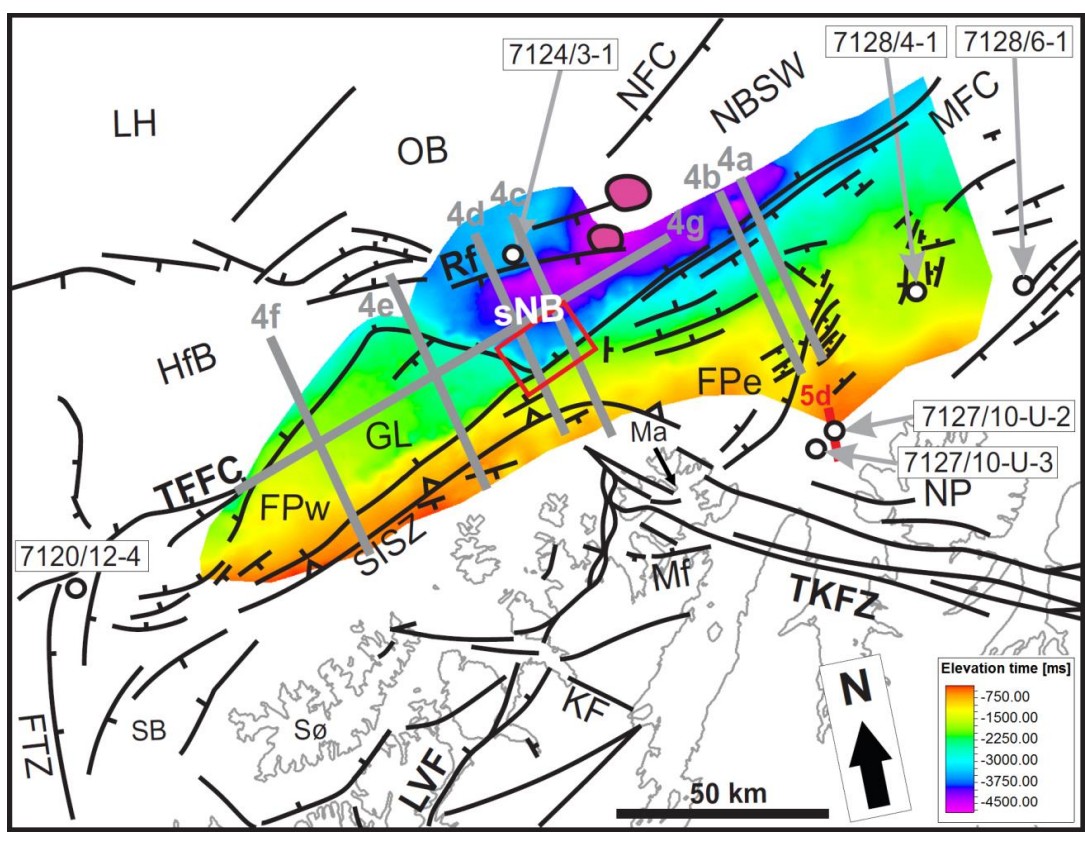

**Figure 2: Regional structural map summarizing the architecture of the Finnmark Platform east (FPe) and west (FPw) and of the southwesternmost Nordkapp basin (sNB). The figure includes a time map of the interpreted mid-Carboniferous reflection. Grey lines show the location of seismic profiles displayed in Figure 4a-g, the red line displays the location of the seismic section shown in Figure 5d and the red frame indicates the location of seismic Z-slices described in Figure 7. White dots show the location of exploration wells and shallow drill-cores while purple blobs represent major salt diapirs in the southernmost part of the Nordkapp Basin (NBSW). See Figure 1 for abbreviations.**





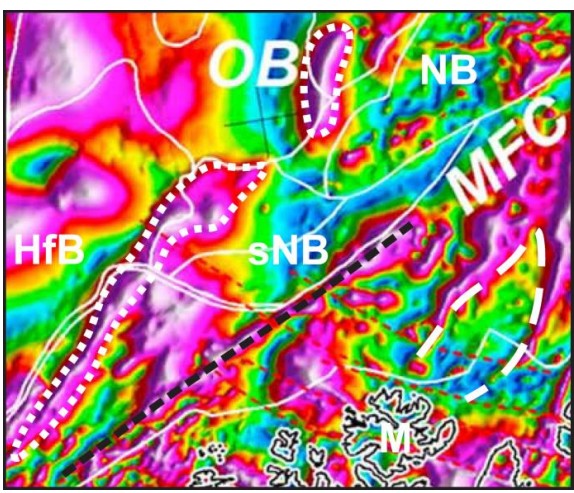

**Figure 3: Zoom in offshore tilt-derivative aeromagnetic data published by Gernigon et al. (2014). The white dashed line on the Finnmark Platform east represents a triangular- to rhomboid-shaped aeromagnetic low that coincides with a Carboniferous basin bounded by zig-zag-shaped brittle faults (e.g. LVF). The dotted white lines on the Finnmark Platform west and on the northern flank of the southwesternmost basin represent ENE-WSW to NE-SW trending ridges of magnetic basement rocks. The dashed black line represents a linear, NE-SW trending, high positive aeromagnetic anomaly that has been tied to the occurrence of the main segment of the MFC (cf. Indrevær et al. 2013). See Figure 1 for abbreviations.**






**Figure 4: Examples of interpreted seismic profiles from the BSS-01 survey (2D) which locations are displayed in Figure 2. Brittle faults are shown in black and depth is in seconds (s) TWT. See Figure 1 for abbreviations; a) Interpreted seismic section that shows a system of Carboniferous horst-graben structures on the Finnmark Platform east; b) Seismic profile showing increased normal displacement across the NW-dipping LVF compared with (a) and thickening of the Carboniferous sedimentary succession within the graben bounded by the LVF. Note the insignificant amount of the displacement accommodated by the northern segment of the MFC in (a) and (b). Black arrows mark brittle faults that bound a triangular-shaped, negative aeromagnetic anomaly (cf. dashed white line in Figure 3); c) Seismic profile showing a highly thickened Carboniferous succession and potential Devonian-lower Carboniferous sedimentary rocks in the southwesternmost Nordkapp basin. Note the large offset accommodated by the main segment of the MFC and the peculiar "U" shape of the southwesternmost Nordkapp basin. Also displayed is a lateral projection of exploration well 7124/3-1; d) Interpreted seismic section that shows the listric geometries of the main segment of the MFC and of the Rolvsøya fault; e) Seismic section showing potential Devonian sedimentary rocks deposited in a NE-SW trending graben above a set of minor, SE-dipping shear zones on the Finnmark Platform west; f) Seismic section showing the listric geometries of the TFFC and**





**MFC, which both seem to sole into the SISZ; g) NE-SW trending seismic cross-section across the Finnmark Platform west and the southwesternmost Nordkapp basin showing the gentle dip of the SISZ to the northeast and a gradual thinning of the upper Carboniferous sedimentary succession towards the southwest. A major NNE-SSW trending, SE-dipping brittle fault seems to offset the SISZ and an intra-basement reflection on the Finnmark Platform west before being truncated by the mid-Carboniferous reflection. The vertical red arrow shows the location of the imaginary prolongation of the TKFZ on the Finnmark Platform west as a comparison with the actual location of the WNW-ESE trending fault segment of the TFFC, which are separated by a distance of ca. 23 km.**



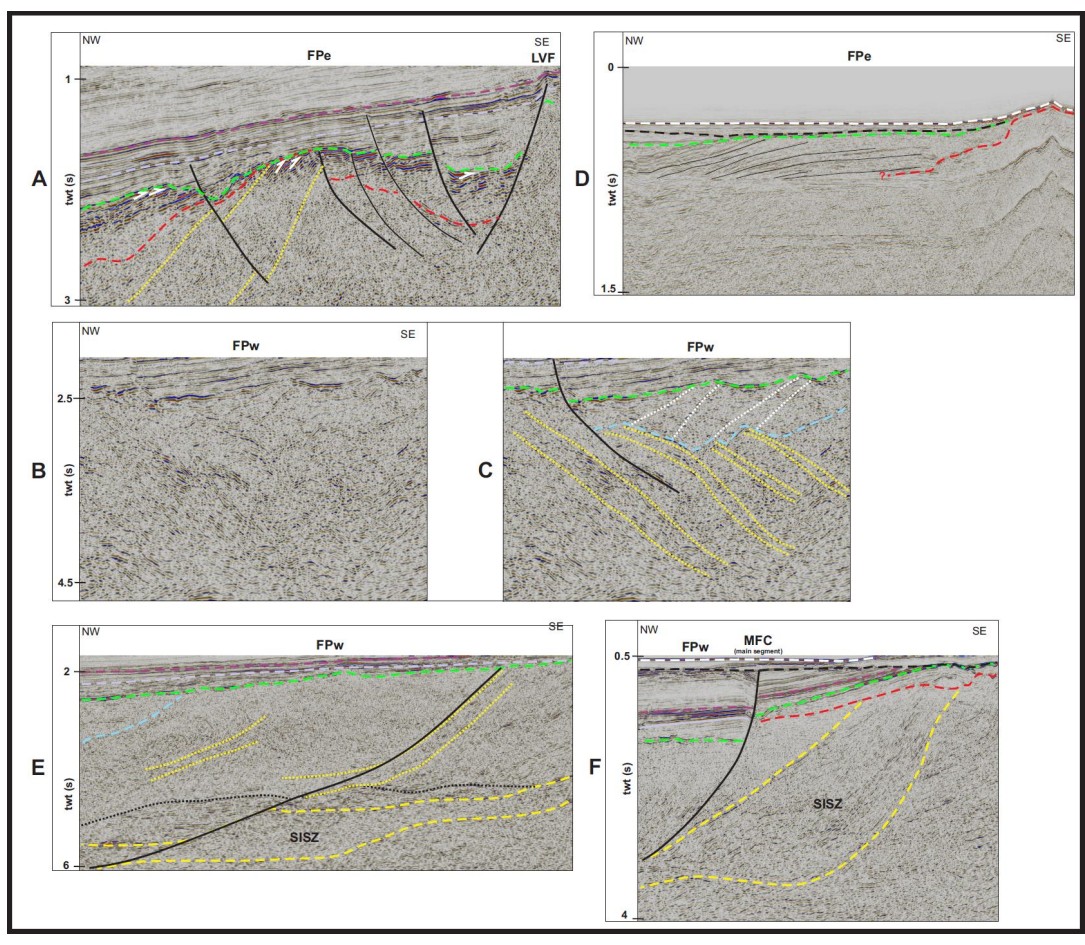

Figure 5: Zoom in seismic sections on the Finnmark Platform east and west. The locations of (a), (b), (c), (e), & (f) are displayed as white frames in Figure 4 and the location of (d) is shown as a red line in Figure 2. See Figure 1 for abbreviations and Figure 4 for seismic reflection legend; a) Interpreted seismic section across the Finnmark Platform east. White arrows represent high-amplitude lower Carboniferous and basement seismic reflections that are truncated upwards (toplaps) by the mid-Carboniferous reflection. Note the contrast between low-amplitude upper Carboniferous-Permian reflections, gently dipping, high-amplitude lower Carboniferous reflections and steeply dipping, high-amplitude basement reflections that possibly belong to a basement-seated shear zone (yellow dotted lines); b) uninterpreted and c) interpreted seismic zoom of a section across presumed Devonian sedimentary rocks and SE-dipping basement shear zones (yellow dotted lines) on the Finnmark Platform west; d) Interpreted seismic section from the IKU-87-BA (2D) survey showing a thick lower Carboniferous succession made up of large clinoforms (thin black lines) on the Finnmark Platform east (location in Figure 2). Note the presence of seismic artifacts in the southeast, including several multiples and NW-dipping diffraction rays; e) Interpreted seismic section across the Finnmark Platform west that displays NE-dipping basement shear zones (yellow dotted lines) including the SISZ (yellow dashed lines); f) Seismic zoom in the SISZ in the footwall of the main segment of the MFC on the Finnmark Platform west. The SISZ is composed of NW-dipping, moderate- to high-amplitude reflections that dip more gently than the MFC but that are steeper than basement reflections in the southeast. Note the significant thickness variations of the SISZ, thick in the footwall of the MFC and thin below the MFC.









**Figure 6: Time surface map of the top reflection of the SISZ and major brittle faults in the SW Barents Sea. Note the spoon-shaped depression formed by the SISZ on the Finnmark Platform west and southwesternmost Nordkapp basin, the abrupt change to a northeastward dip on the Finnmark Platform east, and the two narrow, NE-SW and ENE-WSW trending ridges in the footwall of the TFFC and of the Rolvsøya fault.**






**Figure 7: (a) Intra-Permian seismic time-slice within 3D seismic survey MC3D-MFZ02 in the southwesternmost Nordkapp basin. Dashed black lines correspond to interpreted brittle faults; (b) Seismic time-slice within 3D seismic survey MC3D-MFZ02 near the interpreted mid-Carboniferous reflection in the southwesternmost Nordkapp basin. Black dashed lines represent interpreted brittle faults. See Figure 2 for location.**






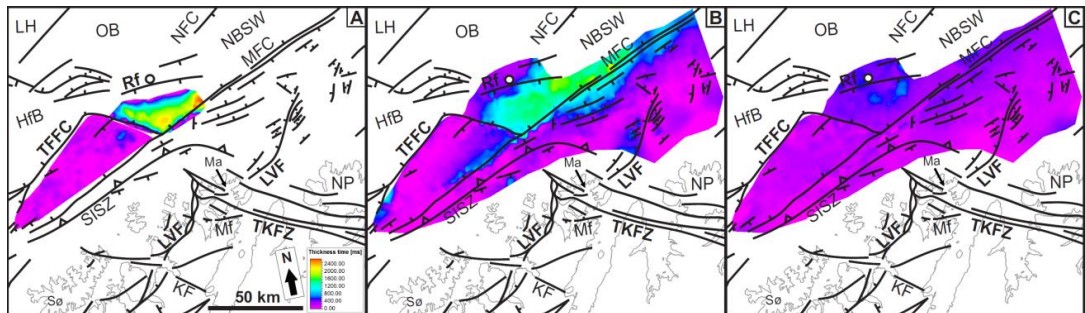

**Figure 8: Thickness maps in milliseconds (ms) two-way time (TWT) of late Paleozoic sedimentary successions on the Finnmark Platform and in the southwesternmost Nordkapp basin. Color scale in (a); a) Thickness map of the Devonian-lower Carboniferous succession on the Finnmark Platform west and in the southwesternmost Nordkapp basin. The succession is thickest in the southwesternmost Nordkapp basin and represents the thickest sedimentary unit of the basin. On the Finnmark Platform west, lower Carboniferous sedimentary rocks are missing but Devonian sedimentary deposits are possibly preserved in an ENE-WSW trending graben adjacent to the southwesternmost Nordkapp basin and bounded to the southeast by the MFC; b) Thickness map of the upper Carboniferous sedimentary succession showing gradual thickening of upper Carboniferous sedimentary rocks in the southwesternmost Nordkapp basin, on the Finnmark Platform west in the hanging-wall of the MFC, and on the Finnmark Platform east in the hanging-wall of the LVF and of a SE-dipping fault that parallels the MFC; c) Thickness map of the Permian succession showing very thin Permian sedimentary deposits and very mild thickness variations within the Permian sedimentary succession throughout the study area.**





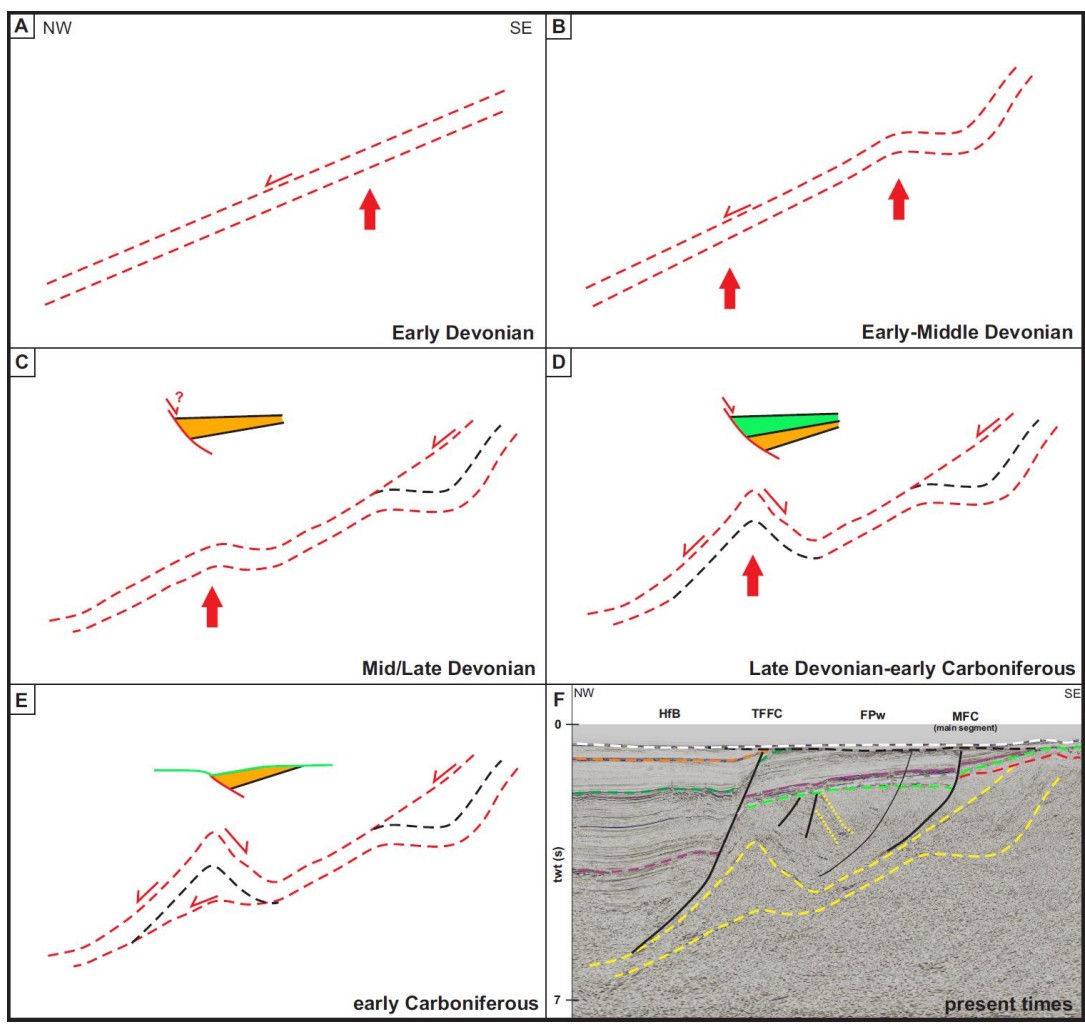

**Figure 9: Evolutionary model that tentatively explains thickness variations along the SISZ. Note that the timing of (a) to (e) is tentative. Dashed red lines in (a) to (e) correspond to tectonically active portions of the SISZ whereas dashed black lines show inactive portions. Red lines in (c), (d) & (e) show presumed normal faults. Thick vertical red arrows indicate exhumation of basement rocks along the SISZ. The model is adapted to the geometry of the SISZ observed below the Finnmark Platform west (see f); a) Extensional reactivation (thin red arrow) of the SISZ in Early Devonian times. Rapid crustal thinning along the upper part of the SISZ triggers exhumation of basement rocks near the coasts of NW Finnmark (thick red arrow); b) In the Early-Middle Devonian, continued extension, crustal thinning and basement exhumation led the upper part of the SISZ to bow. Further crustal thinning triggers exhumation of basement rocks along lower portions of the SISZ (left-hand side, thick red arrow); c) In Mid/Late Devonian times, bowed portions of the SISZ become inactive and excisement (i.e. upwards splaying; cf. Lister & Davis 1989) of the SISZ into its hanging-wall leads to thickening of the upper portion of the SISZ. Continued crustal thinning triggers bending of the lower part of the SISZ (thick red arrow) above which brittle normal faults may have formed and localized the deposition of Devonian sedimentary deposits (orange); d) Further exhumation of basement rocks along lower portions of the SISZ in the Late Devonian-early Carboniferous leads to extreme bending of the SISZ, to antithetic top-to-the-SE extensional faulting and to early Carboniferous syn-tectonic sedimentation (green); e) Towards the end of the early Carboniferous, the lower portion of the SISZ is thickened due to incisement (i.e. downward splaying; cf. Lister & Davis 1989) of the SISZ into bow-shaped portions in its footwall. Core complex exhumation ceased in the Serpukhovian and a major sea-level fall exposed the Finnmark Platform to continental erosion (green line representing the mid-Carboniferous reflection); f) Present times seismic expression of thickness variations along the SISZ (dashed yellow) on the Finnmark Platform west. See Figure 4 for seismic reflections color schemes.**



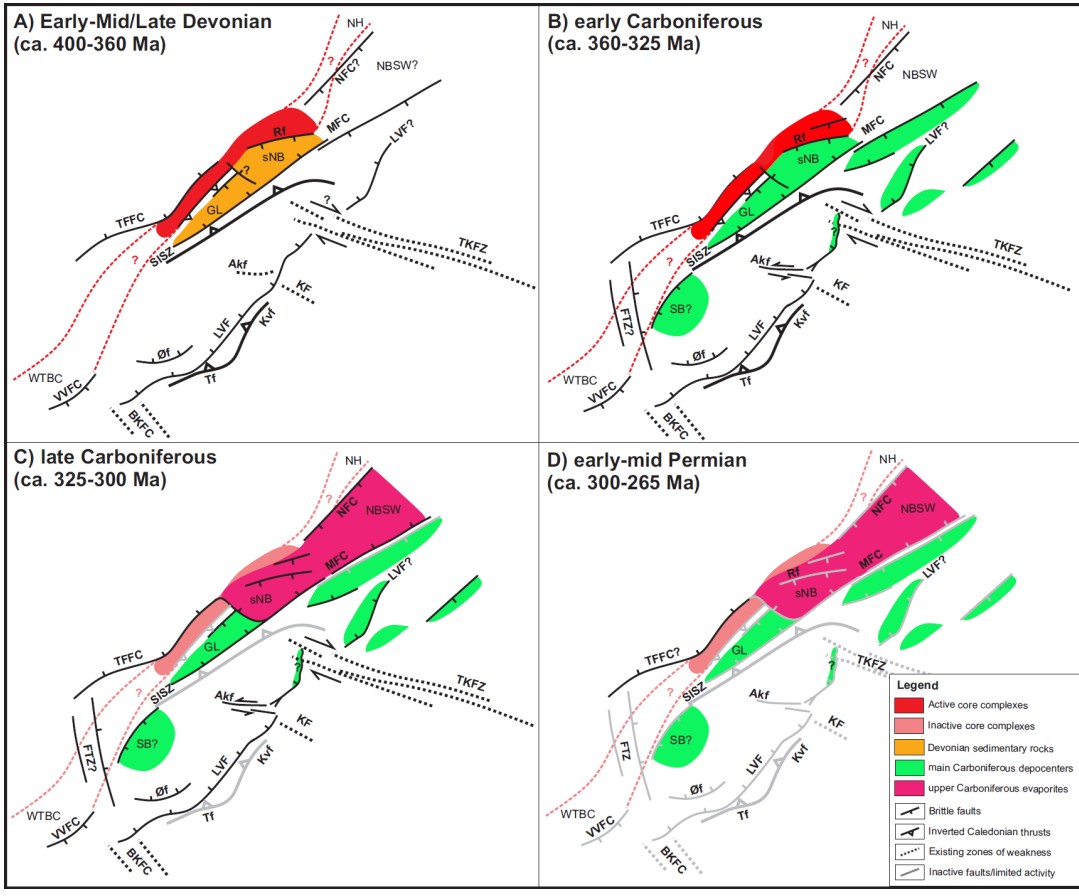

**Figure 10: Map-view figures summarizing the late Paleozoic tectono-sedimentary evolution of the Finnmark Platform and southwesternmost Nordkapp basin (sNB). The tectonic evolution of onshore and nearshore faults in NW Finnmark is from Koehl et al. (submitted). Abbreviations as in Figure 1. a) In the Early to Mid/Late Devonian, major Caledonian thrusts (e.g. SISZ) were inverted as low-angle extensional shear zones and exhumed metamorphic core complexes in the footwall of the TFFC and of the Rolvsøya fault. Thick Devonian sedimentary rocks were deposited within a spoon-shaped trough created by the geometry of the SISZ; b) Core complex exhumation continued through the early Carboniferous, though mostly accommodated by high-angle normal faults, which formed as brittle splays along Caledonian thrusts and shear zones (e.g. MFC, TFFC, Rolvsøya fault and LVF). Core complex exhumation ceased by the end of the Serpukhovian and the WNW-ESE trending fault segment of the TFFC formed as an accommodation cross-fault that decoupled the Finnmark Platform west from the southwesternmost Nordkapp basin, thus contributing to preserve thick Devonian and lower Carboniferous sedimentary successions in the southwesternmost Nordkapp basin while these sedimentary rocks were almost completely eroded on the Finnmark Platform west. Minor graben and half-graben structures formed on the Finnmark Platform east. Precambrian, WNW-ESE to NNW-SSE trending fault zones such as the TKFZ segmented the margin and acted as minor transfer faults that accommodated limited amount of lateral displacement. Lateral movements along these faults ceased in the early Carboniferous; c) In the late Carboniferous, inverted Caledonian thrusts and shear zones became inactive and were truncated by high-angle splay-faults that accommodated the deposition of syn-tectonic sedimentary wedges on the Finnmark Platform east 0and west, and of thick, partly evaporitic deposits in the southwesternmost Nordkapp basin; d) By the end of the Carboniferous, active brittle faulting came to a halt and the Finnmark Platform and the southwesternmost Nordkapp basin are believed to have remained tectonically quiet.**



