# Peer review of "Mid/Late Devonian-Carboniferous collapse basins on the Finnmark Platform and in the southwesternmost Nordkapp basin, SW Barents Sea"

_Solid Earth, 2017_

## Referee Comment (RC1) · T. Phillips (Referee) · 18 Dec 2017

Thank you for the invitation to review this paper by Koehl et al, "Mid/Late Devonian-Carboniferous collapse basins on the Finnmark Platform and in the southwesternmost Nordkapp basin, SW Barents Sea," and for the opportunity to be involved in the open review process.

This manuscript provides a comprehensive overview of how structural inheritance may have played a role in the development of the Finnmark Platform area.

[Figure]

The authors map the 3D geometry of a newly-identified shear zone located on the Finnmark Platform and analyse how this shear zone has influenced the physiography of the overlying rift system. They relate activity along and reactivation of the identified shear zones to the formation of basement ridges and supra-detachment basins, linking these to previously documented examples in the North Sea. Furthermore, the authors propose a new model for the offshore continuation of the Trollfjord-Komagelv Fault Zone, purporting that it does not extend offshore as previously proposed.

The authors present a detailed analysis and seismic interpretation of the main structural elements of the rift system and also the shear zone located within basement, which is underpinned by a comprehensive geological history of the area, both onshore and offshore. This paper provides a very interesting and novel example of how pre-existing shear zones may influence faults and rift systems.

I have a number of issues regarding specific aspects of the paper, as outlined below. I recommend that, following these changes, this paper be accepted for publication in Solid Earth.

The authors state that they identify a NE-SW trending "zone of weakness" on seismic reflection data (LINE 200). Based on the seismic data alone, no inference can be made as to the lithological properties of the structure, rather; what is imaged is a package of prominent inclined reflectivity.

As this reflection package does not directly correlate to any structures as observed onshore, more evidence is required before the authors can state with confidence that this represents a shear zone or a zone of weakness.

In addition, the authors state that "km-thick layers bearing strong basement fabrics. . ." may be resolvable at seismic scale (LINE 438-443). References to shear zones as previously imaged and modelled in seismic data need to be included at this point to back up the, in my view correct, interpretation that this reflection package represents a shear zone. Such references include : Phillips et al. (2016); Reeve et al. (2013);

Fountain et al. (1984)

LINE 456-457 – Can you speculate as to what the minor mylonites and shear zones may correspond to? Could they correspond to fabrics within Caledonian allochthons? Or potentially thrusts between allochthons?

The authors propose a model of core complex exhumation along with excisement and incisement to explain the bowed portions of the SISZ and the exhumation of basement ridges (i.e. Figure 9; Section 5.4).

Whilst I agree that the faults appear to merge down with the shear zone structures at depth, what remains unclear is the mechanism by which the bowed portion of the SISZ forms at deeper levels. What causes the SISZ, which then influences faults in the overlying sedimentary sequence, to be uplifted and bow at a particular location at depth? During core complex exhumation, bowed portions would be expected to form towards the surface, but I am unsure as to what would drive the uplift at deeper levels (i.e. red arrow in Figure 9b, c)

Would it be possible that the fault forms first leading to the passive uplift of the shear zone in its footwall? A more detailed description of this mechanism is required, potentially with more detailed applied to figure 9.

LINE 570-571 – I think that you need to first confirm that the observed changes in thickness along the structure are real and not related to variable imaging quality of the shear zone along strike and at depth. For example, the mylonites/fabrics generating the reflections may destructively interfere in some instances.

More information is required on the data used in this study and the coverage provided (LINE 404). What is the data coverage across the area, which areas are covered by 3D seismic data? What is the typical spacing between 2D lines?

LINE 316-319 – does this imply that the faulting pre-dates the dyke emplacement, or is this able to provide any constraints on the exact dating of the faulting? It needs to

be made clearer if these dykes are associated with the faulting or just place an upper bound on the age of dyke emplacement.

Additional, more minor comments are outlined below

Figure 1 appears very cluttered, with a large number of structural elements labelled on the same figure. As such it can often be difficult to identify specific figures referred to in the text (i.e. the locations of the star symbols, LINE 364; Lofoten-Vesteralen margin, LINE 285). In addition, it is difficult to distinguish between those structures that are fundamental to the text and analysed in detail from more minor structures. Perhaps it would be worth distinguishing the key structural elements. Furthermore, the southwesternmost Nordkapp basin and the area focussed on in the study could be outlined to draw the readers attention.

The regional map shown in figure 1 currently offers little information. This should be changed to a slightly more regional version of that shown in 1A (i.e. northern Norway), allowing some regional structures to be labelled on this map instead.

Figure 2 – Would it be possible to show the location of this figure on Figure 1

Figure 4 – Details of the seismic sections are not clear on both printed and online versions of the manuscript, making it difficult to identify some of the interpretations made in the text. Would benefit from being split over two pages with each section made larger.

Figure 5 – These sections appear better quality than those shown in figure 4, with structures and interpretations clearly visible. However, sections in this figure would still benefit from being made larger.

Figure 5c – the relationship between the shear zones and the later rift-related faults shown here appear similar to the exploitative fault interactions of Phillips et al 2016, where we suggest that the fault exploit mechanical anisotropies represented by the mylonitic layers. Also applicable to LINE 725-729.

Figure 8 – Label each of the individual isochrons with the stratigraphic interval.

Figure 10 - The different shades of red used in the figure can be difficult to make out. LINE 1824 (Figure caption) – spelling mistake "0and"

LINE 49-51 – Sentence doesn't make grammatical sense as it stands currently

LINE 69-70 – the authors state the Senja Shear Zone and the Fugloya Transfer Zone parallel the Trollfjord-Komagelv Fault Zone, this does not appear to be the case in Figure 1, with the SSZ and FTZ appearing almost perpendicular to the TKFZ.

LINE 155 – what differentiates between previous studies that map the TKFZ as a discrete structure and this study, where it is mapped as a series of discrete strands?

LINE 427 – The dykes mentioned are not shown in the magnetic map shown in figure 3

LINE 455-461 – It may be useful to compare with the seismic facies observations of Fazlikhani et al. 2017 based on observations from the northern North Sea.

LINE 469 – Spelling of occasional

LINE 463-467 – I am unable to make out such seismic stratigraphic relationships due to the imaging of the seismic sections shown in figure 4.

LINE 563 – Clarify whether you mean 'curved', in map view or in cross-section?

LINE 575 – I'm slightly confused by this statement, it seems that the causation should be the opposite way around. The correct phrasing and causation is given on LINE 635. The way it is phrased currently implies that the SISZ merges with the TFFC rather than the later-formed TFFC merging with the pre-existing SISZ?

LINE 588 – noteworthy needs to be changed to notably

LINE 609-615 – Good interpretation of the relationship between the two.

LINE 862-870 – Also link to additional examples earlier on to add more weight to the

interpretation of the reflection package as a shear zone

---

## Referee Comment (RC2) · D. Marin (Referee) · 6 Jan 2018

This manuscript presents an interesting interpretation and discussion about the Sørøya-Ingøya Shear Zone and its contribution to the evolution of the basins and fault complexes in the SW Barents Sea during the late Paleozoic. It also discusses the interaction of the different fault complexes in the area. The authors make an extensive review of the literature and use seismic data, published aeromagnetic data and the few wells available to support their interpretations. The English is in general fine, with few spelling mistakes. However, the paper needs to be more concise and organized. The

manuscript and some of the figures need major corrections before it can be published.

Specific comments:

1) Although the manuscript is a good contribution to the understanding of the geology of the SW Barents Sea, it lacks a more global impact. Why should researchers that do not work in the SW Barents Sea read this paper? You can include a paragraph that highlights this issue. But please be concise, as the paper is already very long.

2) The length of the paper should be substantially reduced. Very few people will read the entire paper with such length. In order to do this, I have the following suggestions: -Avoid repetition: you mention three times that the easternmost Hammerfest Basin should be renamed southwesternmost Nordkapp basin, and at least three times you discuss the origin of the Serpukhovian unconformity. Just mention these things once and proceed to the point. -Geological setting: You can reduce this section considerably if you only include what is relevant for your study. A figure with a stratigraphic column could probably help you summarize sections 2.1 and 2.3 into a single paragraph. The geological setting is a little unorganized. For example, in section 2.1.1 you are writing about Precambrian rocks, but suddenly you start describing faults (Lines 155–166, page 6). I suggest that you divide the geological setting in section 2.1, where you only write about lithology, and section 2.2 where you can write about the structural geology. Lines 199–200, page 7 do not belong to the geological setting, it is part of the results. Lines 233–236, page 8; 273–278, pages 9–10 do not belong to geological setting. -Section 5.6 is a summary of what you already have said. You can consider removing this section.

3) Methods: be more specific about the description of your well-seismic tie. Did you make a synthetic seismogram? Which parameters did you use?

4) You need to clarify the meaning of your seismic unit's tops: are they sequence boundaries (if so, what type?), formations tops or just key reflectors with stratigraphic meaning? Because in your results you sometimes write about groups, sequences or

ages. Be consistent and do not mix nomenclatures. In the figure of the stratigraphic column, you can also add your seismic unit's tops.

5) Results: descriptions and interpretations are mixed. You can split each section of the results into a description and an interpretation part, in order to make the results chapter easier to read. And please try also to summarize this section.

6) Discussion: I have a problem with your alternative interpretation of the TKFZ. First you said that the TKFZ dies out before the Finnmark Platform (page 1 line 27; page 37, line 1128; page 38, line 1144), but in your alternative (contradictory) interpretation you suggest that the TKFZ could have been partially eroded in the Finnmark Platform, but it might be possible to find its prolongation in the Loppa or even in the Veslemøy High. To support your interpretation, you mention some WNW–ESE faults in Veslemøy High, referring to Kairanov et al., 2016. First, the figures of this reference are not easy to find for the readers (since this was a conference presentation). Second, what is the timing of the faults in the Veslemøy High compared to the TKFZ? Are they even the same type of faults? You are not showing data that supports your alternative interpretation of the propagation of the TKFZ to the W.

7) Figure 1: the font size of your abbreviations is different. Why do the BSFC and BKFC have a bigger font? Why are the TKFZ and TFFC abbreviations bold?

8) Figure 2: some of your fonts are bold. Why?

9) Figure 3: this figure does not have scale or coordinates. It does not have a color scale. What is the meaning of the red dotted line?

10) Figure 4: seismic sections are very small, and it is very difficult to see any details (e.g seismic character, amplitude, geometries). You need to make them bigger. It is difficult to agree with your descriptions and interpretations if I can not properly see the data. The sections do not have horizontal scale. You should provide the uninterpreted seismic lines (this can be in supplementary material, if there are any restriction on the

number of figures). Sometimes you do not interpret the tops in the entire seismic line. Why? Is it because there is a lot of uncertainty (in that case you could use question marks).

11) Figure 5: fix the order of the figures. After A comes B, not D. Add horizontal scales.

12) Figure 7: it shows a time slice near the mid-Carboniferous. That applies probably only for the hanging wall. Add scale.

13) Figure 8a shows a thickness map of the Devonian–lower Carboniferous, including areas as the sNB. In the seismic lines 4c, d neither the base of the Devonian or the basement are interpreted. How did you make this thickness map? Which reflectors did you use? The Mid- Carboniferous and the SISZ? Also try to make these maps bigger.

14) There are many paragraphs that need a figure as a reference. If not, they are difficult to understand or visualize, ex: page 3, line 69; page 9, line 250; page 10, line 301; page 38, line 1159.

15) Some sentences are very long, for example: page 3, lines 67–73; page 12, lines 360–364; pages 16–17, lines 490–498. Try to split them to make the paper easier to read.

16) Be consistent between the names that you use in the text and the figures. Is the Senje fracture zone in line 285, the same as the Senje Shear Zone in figure 1? Page 19, line 576 says basement highs, but in figure 1 it says basement ridges.

Good luck with the revision.

Dora Marin

---

## Author Comment (AC1) · 31 Jan 2018

Dear Dr. Phillips, Thank you very much for your comprehensive review of our manuscript. Please find our response to your comments organized in three sections: (1) Comments from Dr. Thomas Phillips, (2) Authors' response, (3) Changes implemented. We hope you find it satisfactory and comprehensive.

1. Comments from Thomas Phillips Major comment Comment 1: The authors state that they identify a NE-SW trending "zone of weakness" on seismic reflection data (LINE

200). Based on the seismic data alone, no inference can be made as to the lithological properties of the structure, rather; what is imaged is a package of prominent inclined reflectivity. As this reflection package does not directly correlate to any structures as observed onshore, more evidence is required before the authors can state with confidence that this represents a shear zone or a zone of weakness. Comment 2: In addition, the authors state that "km-thick layers bearing strong basement fabrics: : :"may be resolvable at seismic scale (LINE 438-443). References to shear zones as previously imaged and modelled in seismic data need to be included at this point to back up the, in my view correct, interpretation that this reflection package represents a shear zone. Such references include : Phillips et al. (2016); Reeve et al. (2013); Fountain et al. (1984). Comment 3: LINE 456-457 – Can you speculate as to what the minor mylonites and shear zones may correspond to? Could they correspond to fabrics within Caledonian allochthons? Or potentially thrusts between allochthons? Comment 4: The authors propose a model of core complex exhumation along with excisement and incisement to explain the bowed portions of the SISZ and the exhumation of basement ridges (i.e. Figure 10; Section 5.4). Whilst I agree that the faults appear to merge down with the shear zone structures at depth, what remains unclear is the mechanism by which the bowed portion of the SISZ forms at deeper levels. What causes the SISZ, which then influences faults in the overlying sedimentary sequence, to be uplifted and bow at a particular location at depth? During core complex exhumation, bowed portions would be expected to form towards the surface, but I am unsure as to what would drive the uplift at deeper level (i.e. red arrow in Figure 10b, c) Would it be possible that the fault forms first leading to the passive uplift of the shear zone in its footwall? A more detailed description of this mechanism is required, potentially with more detailed applied to figure 10. Comment 5: LINE 570-571 – I think that you need to first confirm that the observed changes in thickness along the structure are real and not related to variable imaging quality of the shear zone along strike and at depth. For example, the mylonites/fabrics generating the reflections may destructively interfere in some instances. More information is required on the data used in this study and the coverage

provided (LINE 404). What is the data coverage across the area, which areas are covered by 3D seismic data? What is the typical spacing between 2D lines? Comment 6: LINE 316-319 – does this imply that the faulting pre-dates the dyke emplacement, or is this able to provide any constraints on the exact dating of the faulting? It needs to be made clearer if these dykes are associated with the faulting or just place an upper bound on the age of dyke emplacement.

Minor comments figures Comment 7: Figure 1 appears very cluttered, with a large number of structural elements labelled on the same figure. As such it can often be difficult to identify specific figures referred to in the text (i.e. the locations of the star symbols, LINE 364; Lofoten-Vesteralen margin, LINE 285). In addition, it is difficult to distinguish between those structure that are fundamental to the text and analysed in detail from more minor structures. Perhaps it would be worth distinguishing the key structural elements. Furthermore, the southwesternmost Nordkapp basin and the area focussed on in the study could be outlined to draw the readers attention. Comment 8: The regional map shown in figure 1 currently offers little information. This should be changed to a slightly more regional version of that shown in 1A (i.e. northern Norway), allowing some regional structures to be labelled on this map instead. Comment 9: Figure 2 – Would it be possible to show the location of this figure on Figure 1 (same as orange in comment 8) Comment 10: Figure 5 – Details of the seismic sections are not clear on both printed and online versions of the manuscript, making it difficult to identify some of the interpretations made in the text. Would benefit from being split over two pages with each section made larger. Comment 11: Figure 6 – These sections appear better quality than those shown in figure 5, with structures and interpretations clearly visible. However, sections in this figure would still benefit from being made larger. Comment 12: Figure 6c – the relationship between the shear zones and the later rift-related faults shown here appear similar to the exploitative fault interactions of Phillips et al 2016, where we suggest that the fault exploit mechanical anisotropies represented by the mylonitic layers. Also applicable to LINE 725-729. Comment 13: Figure 9 – Label each of the individual isochrons with the stratigraphic interval. Comment 14:

Figure 11 - The different shades of red used in the figure can be difficult to make out.

Minor comments text Comment 15: LINE 1824 (Figure caption) – spelling mistake "0and" Comment 16: LINE 49-51 – Sentence doesn't make grammatical sense as it stands currently Comment 17: LINE 69-70 – the authors state the Senja Shear Zone and the Fugloya Transfer Zone parallel the Trollfjord-Komagelv Fault Zone, this does not appear to be the case in Figure 1, with the SSZ and FTZ appearing almost perpendicular to the TKFZ. Comment 18: LINE 155 – what differentiates between previous studies that map the TKFZ as a discrete structure and this study, where it is mapped as a series of discrete strands? Comment 19: LINE 427 – The dykes mentioned are not shown in the magnetic map shown in figure 4 Comment 20: LINE 455-461 – It may be useful to compare with the seismic facies observations of Fazlikhani et al. 2017 based on observations from the northern North Sea. Comment 21: LINE 469 – Spelling of occasional Comment 22: LINE 463-467 – I am unable to make out such seismic stratigraphic relationships due to the imaging of the seismic sections shown in figure 5. Comment 23: LINE 563 – Clarify whether you mean 'curved', in map view or in cross-section? Comment 24: LINE 575 – I'm slightly confused by this statement, it seems that the causation should be the opposite way around. The correct phrasing and causation is given on LINE 635. The way it is phrased currently implies that the SISZ merges with the TFFC rather than the later-formed TFFC merging with the preexisting SISZ? Comment 25: LINE 588 – noteworthy needs to be changed to notably Comment 26: LINE 609-615 – Good interpretation of the relationship between the two. Comment 27: LINE 862-870 – Also link to additional examples earlier on to add more weight to the interpretation of the reflection package as a shear zone

2. Author's response Comment 1: we agree with the suggestion of the referee, the sentence should be changed accordingly. Comment 2: agreed with and updated. Comment 3: agreed with and updated. Comment 4: we do not think brittle faults formed first. Instead, we propose that progressive crustal thinning due to extensional reactivation of the SISZ and extensive erosion are the triggering and driving mechanisms

for the bowing of the SISZ. First near surface (figure 10a), and gradually along deeper portions of the SISZ now exhumed to shallower crustal level due to crustal thinning and erosion (figure 10b and c). The location of the bowing is far less obvious because seismic data do not allow to see much deeper than the SISZ but perhaps the bowing localized along pre-existing Paleoproterozoic fabrics/heterogeneities (but too speculative to be included in the paper). We agree though that more information must be provided in the figure (10) caption and in discussion section 5.4. Comment 5: agreed with and added relevant information in Methods chapter. The typical spacing for the 2D survey BSS01 was not provided and is therefore not included in the paper. In addition, thickness variations along the SISZ are based on the interpretation of multiple seismic surveys (not shown in our study) of variable quality (the best being survey BSS01). We agree with the comment of Dr. Phillips in which he mentions "mylonites/fabrics generating the reflections may destructively interfere in some instances". Such phenomenon was actually observed on part of the presented seismic survey (BSS01) but none of these seismic sections is showed in the paper because of the low quality of the SISZ reflections on these sections. We argue that showing such a low-quality section may not add much weight to our argumentation and increase the length of the paper, which is already very long. Comment 6: agree with and updated Comment 7: agreed and adjusted. Comment 8: agreed with and changed. Comment 9: agreed with and updated. Comment 10: agreed with and updated. Comment 11: agreed with and updated. Comment 12: too hard to tell from our seismic data. The fault could be either "exploitative" or "merging" according to the nomenclature used in Phillips et al. (2016). Thus, we would rather leave this out of figure 6c and line 725-729. Comment 13: agreed with and implemented. Comment 14: agreed with and color scheme updated. Comment 15: agreed with and changed. Comment 16: agreed with and changed accordingly. Comment 17: agreed with and re-written. Comment 18: clarified. Comment 19: agreed with and updated. Comment 20: agreed with and updated. Comment 21: agreed with and changed. Comment 22: agreed with and corrected. Comment 23: agreed with. Comment 24: agreed with and changed. Comment 25: agreed with

and changed. Comment 26: agreed with. Comment 27: agreed with and updated (cf. comments 8 and 21).

3. Changes implemented Comment 1: "zone of weakness" in LINE 200 was replaced by "package of [. . .] seismic reflections" as suggested by Dr. Phillips. Comment 2: addition of suggested references "Fountain et al., 1984; Reeve et al., 2013; Phillips et al., 2016". Comment 3: the sentence line 456-457 was updated as follow: "We interpret these pronounced internal fabrics as widespread mylonitic foliation separated by internal thrusts within a large-scale shear zone". Comment 4: emphasized that erosion and extensional reactivation of the SISZ are the triggering factor for the bowing of the SISZ. Modification of the figure caption as follow: "a) Extensional reactivation (thin red arrow) of the SISZ in Early Devonian times. Rapid crustal thinning and possible erosion along the upper part of the SISZ triggers exhumation of basement rocks near the coasts of NW Finnmark (thick red arrow); b) In the Early-Middle Devonian, continued extension and erosion further thin the crust and exhume basement rocks in the footwall of the SISZ, leading the upper part of the SISZ to bow. Incremental crustal thinning due to continued extensional reactivation of the SISZ and continental erosion triggers exhumation of basement rocks along lower portions of the SISZ (left-hand side, thick red arrow); c) In Mid/Late Devonian times, bowed portions of the SISZ become inactive and excisement (i.e. upwards splaying; cf. Lister & Davis 1989) of the SISZ into its hanging-wall leads to thickening of the upper portion of the SISZ. Continued extension and erosion (i.e. crustal thinning) trigger bending of the lower part of the SISZ (thick red arrow) above which brittle normal faults may have formed and localized the deposition of Devonian sedimentary deposits (orange)". In addition, multiple minor text modifications were made from line 1049 to line 1063 to emphasize erosion and extension as the trigger mechanisms for bowing of the SISZ. Comment 5: Methods chapter updated as follow: "The seismic interpretation shown in this study is based on publicly available 2D and 3D data from the DISKOS database, thus providing reasonably tight 2D data coverage. However, only one seismic 3D survey was available in the study area". Addition of the following sentence: "In addition,
we analyzed two time-slice from 3D seismic survey MC3D-MFZ02 to constrain fault interaction in map-view". Comment 6: dolerite dyke provide a minimum estimate of the age of the latest faulting event along the TKFZ. The sentence was updated to "Roberts et al. (1991) and Lippard & Prestvik (1997) presented indirect evidence of early Carboniferous dolerite dykes emplaced along and cementing WNW-ESE trending brittle fault segments of the TKFZ onshore Magerøya, thus providing a minimum estimate for the latest stage of faulting along this fault." Comment 7: we deleted useless abbreviations and adjusted the font of the remaining ones so that important faults and basins appear in bold (e.g. southwesternmost Nordkapp basin - sNB). In addition, we added the different parts of the Norwegian continental shelf we refer to in the regional map, including e.g. Lofoten-Vesterålen. Comment 8: changed regional map into zoom in Norwegian shelf showing western Norway, Lofoten-Vesterålen, the North Sea and the Barents Sea. Comment 9: addition of a dashed black frame showing the location of figure 2 in figure 1. The following sentence was added to the caption of figure 1: "Dashed black frame locates Figure 2". Comment 10: the seismic sections of figure 5 were split into 3 to enlarge each section. Comment 11: enlarged seismic sections of figure 6. Comment 12: no changes. Comment 13: in figure 9, each map was labelled with corresponding stratigraphic interval. Comment 14: in figure 11, light red color (inactive core complexes) was replaced by grey. Comment 15: typo corrected. Comment 16: the sentence was changed into "This suture and possibly related deep-seated shear zones, which accommodated e.g. thrust nappe emplacement during the Caledonian Orogeny, are now covered by late Paleozoic to Cenozoic sedimentary basins that formed during multiple episodes of extension." Comment 17: the phrase was changed to "by margin-oblique, NNW-SSE to WNW-ESE trending transfer fault zones, e.g. Senja Shear Zone and Fugløya transfer zone (Indrevær et al., 2013), which may represent analogs of the onshore, Neoproterozoic, WNW-ESE trending Trollfjord-Komagelv Fault Zone (TKFZ)", thus suppressing the erroneous "sub-parallel" adjective. Comment 18: the sentence was rewritten as follow: "The Timanian Orogeny produced major NW-SE trending folds

(Roberts & Siedlecka, 2002) and WNW-ESE trending fault complexes like the TKFZ (Johnson et al., 1978; Herrevold et al., 2009). The TKFZ was mapped as a narrow, single-segment fault strand all the way along the Kola Peninsula in Russia in the east, where it merges with the Sredni-Rybachi Fault Zone (Roberts et al., 1997; Roberts et al., 2011), to the Barents shelf in the west (Gabrielsen, 1984; Gabrielsen & Færseth, 1989; Gabrielsen et al., 1990; Roberts et al. 2011)". In addition, we added the following sentence to show the reader in which way our study of the TKFZ differs from previous works: "We present an alternative model in which the TKFZ splays into multiple fault segments and dies out between the Varanger Peninsula and the Barents shelf". Comment 19: dolerite dykes added to figure 4 as dotted black lines. In addition, an explanatory sentence was added to the figure caption: "Dolerite dykes intruded along WNW-ESE trending fault segments of the TKFZ are shown by dotted black lines." Comment 20: addition of suggested reference: "Fazlikhani et al., 2017". Comment 21: typo corrected. Comment 22: we agree with the comment and forgot to refer to the appropriate seismic zoom in the Base Devonian reflection. Thus, we added a reference to figure 6b and c. Comment 23: added "in cross-section". Comment 24: sentence changed into "where the listric TFFC merges with the shear zone". Comment 25: "Noteworthy" changed into "Notably". Comment 26: no changes. Comment 27: cf. comments 8 and 21 for changes.

Please also note the supplement to this comment:
https://www.solid-earth-discuss.net/se-2017-124/se-2017-124-AC1-supplement.pdf
* * *

---

## Author Comment (AC2) · 31 Jan 2018

Dear Dr. Marin, Thank you very much for your well-organized and comprehensive review of our manuscript. Please find our response to your comments organized in three sections: (1) Comments from Dr. Dora Marin, (2) Authors' response, (3) Changes implemented. We hope you find it satisfactory and comprehensive.

1. Comments from Dr. Dora Marin Comment 1: Although the manuscript is a good contribution to the understanding of the geology of the SW Barents Sea, it lacks a

more global impact. Why should researchers that do not work in the SW Barents Sea read this paper? You can include a paragraph that highlights this issue. But please be concise, as the paper is already very long. Comment 2: The length of the paper should be substantially reduced. Very few people will read the entire paper with such length. In order to do this, I have the following suggestions: -Avoid repetition: you mention three times that the easternmost Hammerfest Basin should be renamed southwesternmost Nordkapp basin, and at least three times you discuss the origin of the Serpukhovian unconformity. Just mention these things once and proceed to the point. -Geological setting: You can reduce this section considerably if you only include what is relevant for your study. A figure with a stratigraphic column could probably help you summarize sections 2.1 and 2.3 into a single paragraph. The geological setting is a little unorganized. For example, in section 2.1.1 you are writing about Precambrian rocks, but suddenly you start describing faults (Lines 155–166, page 6). I suggest that you divide the geological setting in section 2.1, where you only write about lithology, and section 2.2 where you can write about the structural geology. Lines 199–200, page 7 do not belong to the geological setting, it is part of the results. Lines 233–236, page 8; 273–278, pages 9–10 do not belong to geological setting. -Section 5.6 is a summary of what you already have said. You can consider removing this section. Comment 3: Methods: be more specific about the description of your well-seismic tie. Did you make a synthetic seismogram? Which parameters did you use? Comment 4: You need to clarify the meaning of your seismic unit's tops: are they sequence boundaries (if so, what type?), formations tops or just key reflectors with stratigraphic meaning? Because in your results you sometimes write about groups, sequences or ages. Be consistent and do not mix nomenclatures. In the figure of the stratigraphic column, you can also add your seismic unit's tops. Comment 5: Results: descriptions and interpretations are mixed. You can split each section of the results into a description and an interpretation part, in order to make the results chapter easier to read. And please try also to summarize this section. Comment 6: Discussion: I have a problem with your alternative interpretation of the TKFZ. First you said that the

TKFZ dies out before the Finnmark Platform (page 1 line 27; page 37, line 1128; page 38, line 1144), but in your alternative (contradictory) interpretation you suggest that the TKFZ could have been partially eroded in the Finnmark Platform, but it might be possible to find its prolongation in the Loppa or even in the Veslemøy High. To support your interpretation, you mention some WNW–ESE faults in Veslemøy High, referring to Kairanov et al., 2016. First, the figures of this reference are not easy to find for the readers (since this was a conference presentation). Second, what is the timing of the faults in the Veslemøy High compared to the TKFZ? Are they even the same type of faults? You are not showing data that supports your alternative interpretation of the propagation of the TKFZ to the W. Comment 7: Figure 1: the font size of your abbreviations is different. Why do the BSFC and BKFC have a bigger font? Why are the TKFZ and TFFC abbreviations bold? Comment 8: Figure 2: some of your fonts are bold. Why? Comment 9: Figure 4: this figure does not have scale or coordinates. It does not have a color scale. What is the meaning of the red dotted line? Comment 10: Figure 5: seismic sections are very small, and it is very difficult to see any details (e.g seismic character, amplitude, geometries). You need to make them bigger. It is difficult to agree with your descriptions and interpretations if I can not properly see the data. The sections do not have horizontal scale. You should provide the uninterpreted seismic lines (this can be in supplementary material, if there are any restriction on the number of figures). Sometimes you do not interpret the tops in the entire seismic line. Why? Is it because there is a lot of uncertainty (in that case you could use question marks). Comment 11: Figure 6: fix the order of the figures. After A comes B, not D. Add horizontal scales. Comment 12: Figure 8: it shows a time slice near the mid-Carboniferous. That applies probably only for the hanging wall. Add scale. Comment 13: Figure 9a shows a thickness map of the Devonian–lower Carboniferous, including areas as the sNB. In the seismic lines 5c, d neither the base of the Devonian or the basement are interpreted. How did you make this thickness map? Which reflectors did you use? The Mid- Carboniferous and the SISZ? Also try to make these maps bigger. Comment 14: There are many paragraphs that need a figure as a reference. If not, they are difficult to understand or visualize, ex: page 3, line 69; page 9, line 250; page 10, line 301; page 38, line 1159. Comment 15: Some sentences are very long, for example: page 3, lines 67–73; page 12, lines 360–364; pages 16–17, lines 490–498. Try to split them to make the paper easier to read. Comment 16: Be consistent between the names that you use in the text and the figures. Is the Senje fracture zone in line 285, the same as the Senje Shear Zone in figure 1? Page 19, line 576 says basement highs, but in figure 1 it says basement ridges.

2. Authors' response Comment 1: agreed with and added appropriate phrase to the Introduction chapter. Comment 2: -Avoid repetitions: deleted one sentence referring to the change of name of the easternmost Hammerfest basin. Shortened sentence line 899. Deletion of sentence line 899-901. Deletion of sentence line 1087-1091 and addition of the following phrase to the previous sentence ", and in agreement with eustatic sea-level fluctuations at that time (Saunders & Ramsbottom, 1986)". -Geological setting: we agree this section should be updated, including the addition of a simplified stratigraphic chart. The geological setting chapter, though relatively long, is organized chronologically. First, we approach Precambrian basement rocks, then Precambrian faults (e.g. TKFZ; lines 155–166, page 6). Second, we address Caledonian nappe thrusting in North Norway and, third, we review existing studies about post-Caledonian sedimentary basins and faults. We believe it is important to address Precambrian faults (e.g. TKFZ) together with associated rocks and deformation events to indicate that these faults correspond to long-lived, basement-seated faults that may have experienced several episodes of reactivation. Thus, we would prefer to keep the geological setting organized as it is now (chronological order) rather than to split it into lithology and structural geology as suggested. Nonetheless, we understand that the length of the geological setting chapter may partly impact negatively the manuscript and we have proceeding to a partial shortening of this chapter. -Section 5.6: we agree that this section repeats what has already been argued for in previous discussion chapters. However, we believe that this section is essential to our contribution since it links all the faults and basins addressed in previous discussion chapters by providing a chronological evolution of the study area. We would therefore prefer to keep section 5.6. Comment 3: agreed with and updated. Comment 4: agreed with and mostly addressed with the addition of a stratigraphic chart (figure 3; cf. comment 2). We also restricted the use of the term "sequence" to intra-unit/succession reflections, e.g. dotted white lines in Devonian sedimentary unit in figure 5 & 6. Comment 5: the authors agree that distinguishing description from interpretation is important to keep the manuscript clear for the reader. Dr. Marin, herself, judiciously uses "description" and "interpretation" sub-headings in a recent manuscript (Marin et al., 2017). We, however, feel that adding supplementary sub-headings to our manuscript will only lengthen and segment a text already split in multiple chapters and sub-chapters. We therefore prefer not to use the suggested additional sub-headings. Comment 6: to clarify: the fault-tip process zone model is from Koehl et al. submitted. Our model in the present contribution is that the TKFZ may partly be preserved in pre-Devonian basement rocks and observable on seismic data across basement highs. Indeed, the reference support we use (Kairanov et al., 2016) is from a conference presentation, which makes it difficult but not impossible to the reader to check our argumentation. The faults observed on the Veslemøy High are sub-vertical, WNW-ESE to NW-SE trending and, thus, geometrically similar to the fault segments of the TKFZ. Further, WNW-ESE trending faults on the Veslemøy High (Kairanov et al., 2016) do not propagate into Mesozoic-Cenozoic sediments and are constrained to basement rocks, hence suggesting that they may represent analogs or even the westwards continuation of the Neoproterozoic TKFZ. Comment 7: agreed with and adjusted. Comment 8: agreed with and corrected. Comment 9: agreed with and corrected/updated. Comment 10: agreed with and updated. Comment 11: agreed with and fixed. Comment 12: agree that the figure needs a scale-bar. However, we believe it is no need to specify that "Intra-Permian" in (a) and "Mid-Carboniferous" in (b) refer to the hanging-wall of the TFFC and MFC since we already mention "in the southwesternmost Nordkapp basin" for both (a) and (b). We furthermore argue that changing "in the southwesternmost Nordkapp basin" into "in the hanging-wall of the TFFC and MFC" would minimize the attention of the reader to the footwall portion of the seismic cube, which is actually the most important portion of the figure showing that the inferred linkage between the TFFC and TKFZ probably does not exist. Comment 13: the SISZ and adjacent basin-bounding fault complexes were used as base Devonian. We added an explanatory sentence to the figure caption. Comment 14: agreed with and updated with appropriate references, apart from page 9, line 250 where we believe sufficient figure references were used to highlight specific structures. Comment 15: agreed with and changed. Comment 16: agreed with the lack of consistency. The term Senja Shear Zone shall not be used. Instead, we now consistently use "Senja Shear Belt" for the onshore Precambrian belt and "Senja Fracture Zone" for the offshore prolongation of the Senja Shear Belt. Page 19, line 576, "basement highs" should be changed for more consistency.

3. Changes implemented

Comment 1: we highlight the regional impact of our contribution on Arctic regions as follow: "The goal of this paper is to contribute to the understanding of tectonic and sedimentary processes in the Arctic in the Late Devonian-Carboniferous. To achieve this, we demonstrate the presence of an overall NE-SW trending, NW-dipping, basement-seated, low-angle shear zone on the Finnmark Platform, the Sørøya-Ingøya shear zone (SISZ; Figure 1), and to discuss its role played in shaping the SW Barents Sea margin during late/post-orogenic collapse of the Caledonides in late Paleozoic times and its influence on the formation and evolution of Devonian-Carboniferous collapse basins." Comment 2: -Avoid repetitions: deletion of the following sentence: "This basin was named the "easternmost Hammerfest basin" by Omosanya et al. (2015). We find this name inappropriate and tentatively rename this basin the "southwesternmost Nordkapp basin", as argued for later in the text". -Geological setting: addition of a stratigraphic chart for the study area. Deletion of lines 152-153, 158-160, 212-217, 351-353, 384, 392-397 and 410-415. In addition, we proceeded to partial shortening of the results and discussion chapter as follow: deletion of lines 520-521, 645-647, 971, 1109-1112, 1178-1179 and 1245-1247. -Section 5.6:

no changes. Comment 3: the following sentence from the methods chapter was updated to "The present study uses ties to wells 7120/12-4, 7128/4-1 and 7128/6-1 and 7124/3-1 based on publicly available well data (www.npd.no) and private well-tie seismograms". Well-tie seismogram used in the present study are private data and cannot be published. We hope the explanatory sentence is satisfactory as it is now. Comment 4: addition of a simplified stratigraphic chart of late Paleozoic successions and restricted use of the term "sequence". Comment 5: no changes. Comment 6: the final paragraph of section 5.5 was largely modified and now includes the geometrical similarities of faults on the Veslemøy High and fault segments of the TKFZ: "However, if the TKFZ ever extended westwards, portions of its western prolongation might be preserved in offshore basement highs such as the Loppa and Veslemøy highs (Figure 1). More work is needed on this hypothesis, but a possible insight is the recent observation of subvertical, WNW-ESE trending brittle faults analog to the TKFZ in basement rocks of the Veslemøy High (Kairanov et al., 2016)". Comment 7: bold fonts in figure 1 now correspond to the most important faults and basins dealt with in the present contribution. Font size of BSFC and BKFC are now the same as other structural elements. Comment 8: bold fonts now highlight the main faults and basins dealt with in the paper. Comment 9: addition of an arrow pointing northwards, a color-scale from the original publication, of a scale bar and of an explanatory sentence regarding dashed red lines (from the original publication; Gernigon et al. 2014) in the figure caption: "Dashed red lines represent faults inferred by Gernigon et al. (2014)." Comment 10: seismic sections of figure 5 were split to enlarge them and horizontal scale were added. In addition, uninterpreted versions of the sections will be submitted as supplements. Comment 11: order of figures changed as suggested and addition of a scale-bar in (a). Comment 12: scale-bar added to the figure and decapitalizing of "Intra-Permian", which becomes "intra-Permian". Comment 13: added explanatory sentence: "Note that in this part of the margin, the SISZ and basin-bounding faults were used as base Devonian reflections." In addition, the three maps were enlarged as suggested. Comment 14: page 3, line 69, we added a reference to figure 1.
Page 9, line 250, nothing was changed. Page 10, line 301, a reference to figure 1 was added. Page 38, line 1159, reference to figure 1 and Koehl et al. (submitted) were added. Comment 15: Page 3, lines 67–73, the sentence was shortened into "The SW Barents Sea margin off Western Troms and NW Finnmark is segmented by margin-oblique, NNW-SSE to WNW-ESE trending transfer fault zones, e.g. Senja Fracture Zone and Fugløya transfer zone (Indrevær et al., 2013), which may represent analogs of the onshore, Neoproterozoic, WNW-ESE trending Trollfjorden-Komagelva Fault Zone (TKFZ) in eastern Finnmark (Siedlecki, 1980; Herrevold et al., 2009) and to the Kokelv Fault on the Porsanger Peninsula (Figure 1; Gayer et al., 1985; Lippard & Roberts, 1987; Rice, 2013)" and the following sentence was added later in the same paragraph: "Onshore-nearshore, margin-parallel fault complexes include the Langfjord-Vargsund fault (LVF; Figure 1) trending NE-SW and possibly representing an analog to the TFFC and MFC". Page 12, lines 360–364, the sentence was split into two as follow: "Devonian sedimentary rocks are yet to be reported in North Norway and along the SW Barents Sea margin. However, Devonian sedimentary deposits are present in western Norway (Osmundsen & Andersen, 2001) where they represent a several km-thick succession made up with clastic deposits that notably include rhythmic sandstone and coarse-grained conglomerate units. These were deposited in the hanging-wall of a major, low-angle extensional shear zone, the Nordfjord-Sogn Detachment Zone (Séranne et al., 1989; Wilks & Cuthbert, 1994; Osmundsen & Andersen, 2001)". Pages 16–17, lines 490–498, the sentence was shortened and split as follow: "On the Finnmark Platform (Figure 1 & 2), the base of upper Carboniferous sedimentary sequences is difficult to identify (cf. "mid-Carboniferous" reflection in figure 5). In places, it appears as a linear, moderate to low amplitude seismic reflection that separates subparallel reflections of lower and upper Carboniferous sedimentary rocks, whereas in other places this reflection is irregular and truncates high-amplitude coal-bearing sedimentary deposits of the Billefjorden Group, and/or high-amplitude reflections produced by basement rocks (figure 6a), and/or low-amplitude reflections in Devonian sedimentary strata (figure 6b & c)". Comment 16: minor changes include lines 69-70 where "Senja Shear Zone" becomes "Senja Fracture Zone", figure 1 where "SSZ" becomes "SFZ" and line 1705 where "SSB = Sørøy sub-basin" becomes "Senja Shear Belt" and "SFZ = Senja Fracture Zone" was added. In addition, page 19 line 576, "basement highs" was changed into "basement ridges".

Please also note the supplement to this comment:
https://www.solid-earth-discuss.net/se-2017-124/se-2017-124-AC2-supplement.pdf

**Supplement:**

[revised manuscript text omitted]
 by crustal thinning and orogenic collapse through dominated by dominant top-to-the-NW displacement movement along the SISZ. and later e

Exhumation of the SISZ and underlying basement ridges as a metamorphic core complex was probably triggered by extensional reactivation of the SISZ combined to continental erosion, leading to crustal thinning. Reactivation of these exhumed basement ridges occurred by normal faulting along new, steep, brittle faults such as the main segment of the MFC and the NNE-SSW trending fault segment of the TFFC (cf. Figure 5Figure 5Figure 4f), likely due to incisement and excisement processes (Figure 10Figure 10Figure 9; Lister & Davis, 1989). These processes also contributed to a the progressive exhumation of basement rocks as ENE-WSW and NE-SW trending basement ridges along bowed portions of the SISZ (cf. Figure 5Figure 5Figure 4c-f and Figure 11Figure 11Figure 10a & b). We believe that these ridges were part of a larger-scale NE-SW trending metamorphic core complex that included the Norsel High and the two basement ridges located in the footwall of the TFFC and the Rolvsøya fault. Farther south, this core complex may be linked to the West Troms Basement Complex (Bergh et al., 2010) and the Lofoten Ridge (Blystad et al., 1995; Figure 11Figure 11Figure 10). Such a regional link is favored by the alignement of NE-SW trending, high-positive gravimetric anomalies that characterize these ridges (Olesen et al., 2010; Gernigon et al., 2014). The timing of final core complex exhumation can be constrained to Mid/Late Devonian-early Carboniferous and possibly linked to the regional Serpukhovian unconformity on the Finnmark Platform (cf. Figure 5Figure 5Figure 4a, b, e, f & g and Figure 6Figure 6Figure 5a-c; Bugge et al., 1995), in accordance with Sartini-Rideout et al. (2006) and Hallett et al. (2014) in northeast Greenland.

The exhumation of basement ridges as metamorphic core complexes along the inverted SISZ and subsequent normal faulting along the MFC and TFFC created a deep, spoon-shaped topographic depression on the Finnmark Platform west and in the southwesternmost Nordkapp basin (Figure 5Figure 5Figure 4c, d, e & g, Figure 6Figure 6Figure 5b & c and Figure 11Figure 11Figure 10a). These depressions were filled with thick Devonian clastic deposits analog to those observed in Middle Devonian collapse basins in western Norway (Séranne et al., 1989; Osmundsen & Andersen, 2001), and with lower Carboniferous coal-bearing and clastic sedimentary rocks of the Billefjorden Group (Figure 3Figure 3) deposited unconformably above Devonian strata (cf. Figure 11Figure 11Figure 10b). These collapse basins are also likely responsible for the gravimetric low observed on the Finnmark Platform west.: the Gjesvær Low (Figure 4Figure 4Figure 3).

On the Finnmark Platform west, the final stages of core complex exhumation and a major phase of eustatic sea-level fall in the Serpukhovian (Saunders & Ramsbottom, 1986) led to extensive erosion of Devonian and lower Carboniferous sedimentary rocks, therefore explaining the absence of lower Carboniferous sedimentary deposits and the erosional truncation of Devonian sedimentary strata along this part of the margin (Figure 5Figure 5Figure 4e-g & Figure 6Figure 6Figure 5b & c). On the Finnmark Platform east, lower Carboniferous sedimentary rocks are preserved as minor syn-tectonic sedimentary wedges within small triangular grabens and half-grabens that correlate with aeromagnetic lows (dashed white line in Figure 4Figure 4Figure 3).

These grabens are bounded by zigzag-shaped, Late Devonian-Carboniferous normal faults such as the LVF (Figure 5Figure 5Figure 4a & b and Figure 6Figure 6Figure 5a; Koehl et al., submitted), which coincide with narrow, high-positive aeromagnetic anomalies (cf. Figure 4Figure 4Figure 3 and black vertical arrows in Figure 5Figure 5Figure 4a & b). In addition, triangular basins like the graben bounded by the LVF and the southwesternmost Nordkapp basin were partly offset and segmented by WNW-ESE trending transfer faults that accommodated small amount of strike-slip displacementmotion. Examples include the TKFZ onshore NW Finnmark, which may offsets the LVF in a right-lateral fashion (Koehl et al., submitted), and accommodation cross-faults (Sengör, 1987) that accommodated large amount of orogen-parallel extension through normal dip-slip movement, for examplee.g. the WNW-ESE trending fault segment of the TFFC (Figure 5Figure 5Figure 4g and Figure 9Figure 9Figure 8a).

In the late Serpukhovian, a regional episode of eustatic sea-level rise (Saunders & Ramsbottom, 1986) flooded the Finnmark Platform east and west and allowed the deposition of sedimentary rocks of the upper Carboniferous Gipsdalen Group (Figure 3Figure 3). These rocks occur as syn-tectonic sedimentary wedges that thicken in the hanging-wall of basin-bounding normal faults such as the LVF on the Finnmark Platform east and the main segment of the MFC on the Finnmark Platform west (Figure 5Figure 5Figure 4a, b, e & f and Figure 11Figure 11Figure 10c). Similarly, in the southwesternmost Nordkapp basin, which may have remained flooded through the entire phase of eustatic sea-level fall and core complex exhumation, thick, partly evaporitic, upper Carboniferous sedimentary rocks were deposited in the basin and these are thickest at the intersection between of the TFFC and the MFC (Figure 5Figure 5Figure 4c, d & g and Figure 9Figure 9Figure 
[revised manuscript text omitted]

---

## Author Comment (AC3) · 31 Jan 2018

Dear David,

Thank you very much for your input on the manuscript; it is, as always, highly appreciated! Here is our response to your comments. We hope the changes we implemented improve the shortcomings of the manuscript highlighted by your comments and suggestions. Please do not hesitate to contact us shall this not be the case for some comments.

1. Comments from Dr. Roberts

Comment 1: mix-ups of British and American spellings, even in the same paragraph (e.g. p.4 Archaean (Brit) vs Archean (Am).

Comment 2: The spelling of the TKFZ by the way is Trollfjorden-Komagelva Fault Zone.

Comment 3: In Figs. 1 & 2 the positioning of the acronym TKFZ is quite wrong. Put it on the Varanger Peninsula (type locality) in Figure 1. It certainly isn't in Laksefjord. However, there are, as you know, very many TKFZ-parallel faults in this part of northern Finnmark (ref my 250K and 500K map-sheets, and the Lippard/Roberts papers); and I have walked across or along most of them in the late-70s, 80s, early-90s. They are mostly normal faults, with this component of movement likely to be Early Carboniferous (see the Nasuti/Rob/Gern mafic dykes paper).

Comment 4: In Fig. 1 the Nordkapp Basin is spelled incorrectly.

Comment 5: On p. 5 line 146 – there is no such thing as the Tanafjord-Varangerfjord Group.

Comment 6: On line 150 the Timanian foreland basin is in the pericratonic 'Gaissa Basin' realm, not the Barents Sea Group (see Zhang et al. 2015 – attached pdf).

Comment 7: On p.6 under 2.1.2, the idea (Kirkland) that the Kalak strata were exotic and originated on Laurentia, and now lie above an inter-continental suture zone (base of Middle Allochthon) has been shown to be groundless (see e.g. Zhang et al. 2016 – attached pdf). The Kalak rocks are most definitely Baltican (see also my NJG V.87 paper from 2007).

Comment 8: I have always been very sceptical about the notion of late-Caledonian orogenic collapse in Finnmark. This was really dramatic in western & central Norway, following c. 200 km of subduction with eclogites, coesite and microdiamonds, but diminishes in intensity northwards. In Finnmark we have inferred late-Scandian extensional microstructures only on the western flank of the Repparfjord window. In

Porsangerfjord, there are late-Scandian, brittle, ESE-directed contractional structures dated to c. Mid Devonian time. So the sandstones beneath the Carboniferous strata in wellcores are likely to be Late Devonian in age (which fits with the evidence of rifting in Late Dev time in NE Varanger and in large areas of NW Russia).

2. Author's response

Comment 1: agreed, we noticed these inconsistencies and made changes where necessary.

Comment 2: agreed and updated.

Comment 3: agreed with and updated figure. We fully agree with you to say that there are many WNW-ESE trending faults and that they accommodated a presumably early Carboniferous component of normal/strike-slip faulting as shown by the dating of Lippard & Prestvik (1997).

Comment 4: agreed with and corrected.

Comment 5: agreed with and corrected.

Comment 6: agreed with and updated/modified using Siedlecka & Roberts (1992; excursion guidebook) as key reference instead of the suggested Zhang et al. 2015 reference.

Comment 7: we most definitely agree that the following references addressing the provenance of rocks of the Kalak Nappe Complex should be cited in the geological setting chapter of our paper: Roberts 2007, Zhang et al. 2016.

Comment 8: the conclusions of our paper is not incompatible with your comment, i.e. it may still have occurred to a lesser extent than in southern and mid-Norway. There are strong indications of Devonian inversion of basement-seated shear zones in Lofoten-Vesterålen (cf. Steltenpohl et al. 2011) and of Late Devonian-early Carboniferous faulting in Troms (Laksvatn and Vannareid faults; Davids et al., 2013) and Finnmark (Kvenklubben, Markopp and Talvik faults; Torgersen et al. 2014; Koehl et al. 2016). We are also aware of the Middle/Late Devonian extensional event in Russia (Pease et al., 2016), e.g. Kontozero Graben (Kramm et al. 1993) and dolerite dykes (Roberts & Onstott 1995).

3. Changes implemented

Comment 1: line 111, "Archaean" becomes "Archean".

Comment 2: lines 25, 27, 41, 42-43, 71, 1102. 1296 and 1706, "Trollfjord-Komagelv" was changed into "Trollfjorden-Komagelva".

Comment 3: location of "TKFZ" acronym changed to the Varanger Peninsula.

Comment 4: in figure 1, "Norkapp Basin" becomes "Nordkapp Basin".

Comment 5: the sentence erroneously referring to the "Tanafjord-Varangerfjord Group" was modified as follow: "A thin cover of Neoproterozoic to Cambrian (para-) autochthonous metasedimentary rocks occurs on top of Paleoproterozoic basement rocks in Finnmark (Siedlecki, 1980; Ramsay et al., 1985; Andresen et al. 2014; Corfu et al., 2014). Other Neoproterozoic-Ordovician units in eastern Finnmark include metasedimentary rocks of the Barents Sea and Tanafjorden-Varangerfjorden regions (Siedlecki, 1980; Siedlecka & Roberts, 1992) which are exposed on the Varanger Peninsula (Figure 1)." Comment 6: cf. comment 5 for implemented changes.

Comment 7: the sentence referring to the hypothesis of Kirkland et al. (2008) and addressing a potential exotic origin of the Kalak Nappe Complex was updated as follow: "The Kalak Nappe Complex was previously considered to represent an exotic terrane accreted on the Laurentian margin of Rodinia prior to the rifting of the Iapetus Ocean, and to have later been thrusted over Baltica during the Caledonian Orogeny (Kirkland et al., 2008). However, paleocurrent and geochronological data suggest these rocks to be of Baltican origin (Roberts, 2007; Zhang et al., 2016)." In addition, the two references were added to the reference list.

Comment 8: no changes.

Please also note the supplement to this comment:
https://www.solid-earth-discuss.net/se-2017-124/se-2017-124-AC3-supplement.pdf
* * *
[Figure]

**Supplement:**

[revised manuscript text omitted]
 by crustal thinning and orogenic collapse through dominated by dominant top-to-the-NW displacement movement along the SISZ. and later e

Exhumation of the SISZ and underlying basement ridges as a metamorphic core complex was probably triggered by extensional reactivation of the SISZ combined to continental erosion, leading to crustal thinning. Reactivation of these exhumed basement ridges occurred by normal faulting along new, steep, brittle faults such as the main segment of the MFC and the NNE-SSW trending fault segment of the TFFC (cf. Figure 5Figure 5Figure 4f), likely due to incisement and excisement processes (Figure 10Figure 10Figure 9; Lister & Davis, 1989). These processes also contributed to a the progressive exhumation of basement rocks as ENE-WSW and NE-SW trending basement ridges along bowed portions of the SISZ (cf. Figure 5Figure 5Figure 4c-f and Figure 11Figure 11Figure 10a & b). We believe that these ridges were part of a larger-scale NE-SW trending metamorphic core complex that included the Norsel High and the two basement ridges located in the footwall of the TFFC and the Rolvsøya fault. Farther south, this core complex may be linked to the West Troms Basement Complex (Bergh et al., 2010) and the Lofoten Ridge (Blystad et al., 1995; Figure 11Figure 11Figure 10). Such a regional link is favored by the alignement of NE-SW trending, high-positive gravimetric anomalies that characterize these ridges (Olesen et al., 2010; Gernigon et al., 2014). The timing of final core complex exhumation can be constrained to Mid/Late Devonian-early Carboniferous and possibly linked to the regional Serpukhovian unconformity on the Finnmark Platform (cf. Figure 5Figure 5Figure 4a, b, e, f & g and Figure 6Figure 6Figure 5a-c; Bugge et al., 1995), in accordance with Sartini-Rideout et al. (2006) and Hallett et al. (2014) in northeast Greenland.

The exhumation of basement ridges as metamorphic core complexes along the inverted SISZ and subsequent normal faulting along the MFC and TFFC created a deep, spoon-shaped topographic depression on the Finnmark Platform west and in the southwesternmost Nordkapp basin (Figure 5Figure 5Figure 4c, d, e & g, Figure 6Figure 6Figure 5b & c and Figure 11Figure 11Figure 10a). These depressions were filled with thick Devonian clastic deposits analog to those observed in Middle Devonian collapse basins in western Norway (Séranne et al., 1989; Osmundsen & Andersen, 2001), and with lower Carboniferous coal-bearing and clastic sedimentary rocks of the Billefjorden Group (Figure 3Figure 3) deposited unconformably above Devonian strata (cf. Figure 11Figure 11Figure 10b). These collapse basins are also likely responsible for the gravimetric low observed on the Finnmark Platform west.: the Gjesvær Low (Figure 4Figure 4Figure 3).

On the Finnmark Platform west, the final stages of core complex exhumation and a major phase of eustatic sea-level fall in the Serpukhovian (Saunders & Ramsbottom, 1986) led to extensive erosion of Devonian and lower Carboniferous sedimentary rocks, therefore explaining the absence of lower Carboniferous sedimentary deposits and the erosional truncation of Devonian sedimentary strata along this part of the margin (Figure 5Figure 5Figure 4e-g & Figure 6Figure 6Figure 5b & c). On the Finnmark Platform east, lower Carboniferous sedimentary rocks are preserved as minor syn-tectonic sedimentary wedges within small triangular grabens and half-grabens that correlate with aeromagnetic lows (dashed white line in Figure 4Figure 4Figure 3).

These grabens are bounded by zigzag-shaped, Late Devonian-Carboniferous normal faults such as the LVF (Figure 5Figure 5Figure 4a & b and Figure 6Figure 6Figure 5a; Koehl et al., submitted), which coincide with narrow, high-positive aeromagnetic anomalies (cf. Figure 4Figure 4Figure 3 and black vertical arrows in Figure 5Figure 5Figure 4a & b). In addition, triangular basins like the graben bounded by the LVF and the southwesternmost Nordkapp basin were partly offset and segmented by WNW-ESE trending transfer faults that accommodated small amount of strike-slip displacementmotion. Examples include the TKFZ onshore NW Finnmark, which may offsets the LVF in a right-lateral fashion (Koehl et al., submitted), and accommodation cross-faults (Sengör, 1987) that accommodated large amount of orogen-parallel extension through normal dip-slip movement, for examplee.g. the WNW-ESE trending fault segment of the TFFC (Figure 5Figure 5Figure 4g and Figure 9Figure 9Figure 8a).

In the late Serpukhovian, a regional episode of eustatic sea-level rise (Saunders & Ramsbottom, 1986) flooded the Finnmark Platform east and west and allowed the deposition of sedimentary rocks of the upper Carboniferous Gipsdalen Group (Figure 3Figure 3). These rocks occur as syn-tectonic sedimentary wedges that thicken in the hanging-wall of basin-bounding normal faults such as the LVF on the Finnmark Platform east and the main segment of the MFC on the Finnmark Platform west (Figure 5Figure 5Figure 4a, b, e & f and Figure 11Figure 11Figure 10c). Similarly, in the southwesternmost Nordkapp basin, which may have remained flooded through the entire phase of eustatic sea-level fall and core complex exhumation, thick, partly evaporitic, upper Carboniferous sedimentary rocks were deposited in the basin and these are thickest at the intersection between of the TFFC and the MFC (Figure 5Figure 5Figure 4c, d & g and Figure 9Figure 9Figure 
[revised manuscript text omitted]